# VDW-GNNs: Vector diffusion wavelets for geometric graph neural networks

David R Johnson [1]   Alexander Sietsema [2]   Rishabh Anand [3]   Deanna Needell [2]   Smita Krishnaswamy [3 4]
Michael Perlmutter [1 5]

## Abstract

We introduce vector diffusion wavelets (VDWs), a novel family of wavelets inspired by the vector diffusion maps algorithm that was introduced to analyze data lying in the tangent bundle of a Riemannian manifold. We show that these wavelets may be effectively incorporated into a family of geometric graph neural networks, which we refer to as VDW-GNNs. We demonstrate that such networks are effective on synthetic point cloud data, as well as on real-world data derived from wind field and neural activity measurements. Theoretically, we prove that these new wavelets have desirable frame theoretic properties, similar to traditional diffusion wavelets. Additionally, we prove that these wavelets have useful symmetries with respect to rotations and translations.

## 1. Introduction

The field of Geometric Deep Learning (GDL) (Bronstein et al., 2017; 2021) aims to extend the success of deep learning to data sets with geometric structure such as graphs and manifolds. Crucially, most GDL methods aim to represent the data points in a manner that respects the intrinsic structure and symmetries of the data (Borde & Bronstein, 2025). Furthermore, they aim to represent two data points the same way if they differ only by an uninformative deformation, such as the relabeling of the vertices of a graph or a global isometry of a manifold (Li et al., 2025; Wang et al., 2022a; Perlmutter et al., 2020a). *In this paper, we focus on geometric graphs where the vertices lie in Euclidean space $\mathbb{R}^D$, with both scalar- and vector-valued node features.*

[1]Program in Computing, Boise State University, Boise, Idaho, USA [2]Department of Mathematics, UCLA, Los Angeles, CA, USA [3]Department of Computer Science, Yale University, New Haven, CT, USA [4]Department of Genetics, Yale University, New Haven, CT, USA [5]Department of Mathematics, Boise State University, Boise, Idaho, USA. Correspondence to: Michael Perlmutter <mperlmutter@boisestate.edu>.

*Proceedings of the $43^{rd}$ International Conference on Machine Learning*, Seoul, South Korea. PMLR 306, 2026. Copyright 2026 by the author(s).

In some cases, we assume that the vertices lie upon an unknown $d$-dimensional manifold $\mathcal{M}$ with $d < D$. In these cases, we will generally assume that the features take (vector) values in the tangent bundle $\mathcal{T}\mathcal{M}$, and our method will aim to leverage this intrinsic low-dimensional structure. Additionally, when appropriate, we will show that our method has desirable symmetries with respect to rigid motions in the ambient space $\mathbb{R}^D$, following the lead of equivariant GNNs (Satorras et al., 2021; Han et al., 2022; Duval et al., 2024).

Most common GNNs are based on the *message-passing* paradigm in which each node is represented in a manner informed by its immediate neighbors, often by local averaging (Kipf & Welling, 2016; Veličković et al., 2018; Xu et al., 2019; Khemani et al., 2024). This promotes similar representations for adjacent vertices, which is a useful heuristic for, e.g., node-classification tasks on homophilous citation networks. However, after each layer of a simple message passing network, the representation of each node becomes progressively smoother. Therefore, in such networks, the total number of layers must be kept small, typically to two or three, to avoid oversmoothing (Nt & Maehara, 2019; Qureshi et al., 2023), where the node representations become increasingly similar, thus limiting their utility for machine learning tasks unless additional mechanisms or connections are added to the network. On the other hand, since each message-passing layer acts locally, this may create a new problem: *underreaching* (Lu et al., 2024), in which the network struggles with tasks requiring the utilization of long-range interactions.

Several possible solutions to the oversmoothing versus underreaching tradeoff have been proposed in the literature, including the use of residual connections (He et al., 2016) and jumping knowledge mechanisms (Xu et al., 2018). In this paper, we will focus on diffusion wavelets (Coifman & Maggioni, 2006) and *geometric scattering transforms* (Zou & Lerman, 2019; Gama et al., 2018; Gao et al., 2019) as well as associated GNNs (Min et al., 2022; 2021; Tong et al., 2024; Bhaskar et al., 2022; Xu et al., 2023; Viswanath et al., 2024). The use of diffusion wavelets allows such networks to capture multiscale graph geometry in a single layer while avoiding oversmoothing (Wenkel et al., 2022). In this work, we extend diffusion wavelets and associated

neural networks to the setting of vector-valued features. Our primary contributions are:

- We introduce *vector diffusion wavelets* (VDWs) for graphs with vector-valued node features.

- We prove that these wavelets have similar frame bounds to traditional diffusion wavelets and that they are equivariant with respect to rotations in the ambient space.

- We show that VDWs may be effectively incorporated into VDW-GNNs, and demonstrate the utility of VDW-GNNs on synthetic and real-world data sets drawn from wind fields and neural-activity manifolds.

## 2. Background

We let $G = (V, E, w)$ be a weighted, undirected, connected graph with vertices (nodes) $V = \{v_1, \cdots, v_n\}$, edges $E \subseteq \binom{V}{2} = \{\{v_i, v_j\} : i \neq j\}$, and weighting function $w : E \to \mathbb{R}$. We let $\mathbf{A} \in \mathbb{R}^{n \times n}$ denote the weighted adjacency matrix. For most of this work, we will focus on geometric graphs, where the vertices $v_1, \ldots, v_n$ lie in $\mathbb{R}^D$, and we will denote the $i$-th coordinate of the vertex $v_j$ by $v_j[i]$.

Adopting the perspective of *graph signal processing* (GSP) (Shuman et al., 2013; Ortega et al., 2018), we interpret the features associated to each node as signals, i.e., functions defined on the $V$. We assume that we are given $F_{\text{scalar}}$ scalar-valued signals $\mathbf{x}_i : V \to \mathbb{R}$, $1 \leq i \leq F_{\text{scalar}}$, and $F_{\text{vec}}$ vector-valued signals $\mathbf{w}_i : V \to \mathbb{R}^d$, $1 \leq i \leq F_{\text{vec}}$. When convenient, we will identify the signal $\mathbf{x}_i$ with the vector in $\mathbb{R}^n$ defined by $\mathbf{x}_i[j] = \mathbf{x}_i(v_j)$. Additionally, we may organize the scalar-values signals into an $n \times F_{\text{scalar}}$ matrix $\mathbf{X} = \mathbf{X}^{(0)}$, whose $i$-th column is $\mathbf{x}_i$ so that the row $\mathbf{X}[j, :]$ contains the features associated to $v_j$. Similarly, we may organize the vector-valued signals into an $n \times F_{\text{vec}} \times d$ tensor $\mathbf{W} = \mathbf{W}^{(0)}$.

### 2.1. Diffusion Wavelets and Geometric Scattering

The geometric scattering transform (Zou & Lerman, 2019; Gama et al., 2018; Gao et al., 2019) provides an alternative to message-passing, which, as shown in Wenkel et al. (2022), allows one to circumvent the oversmoothing-versus-underreaching tradeoff via the use of diffusion wavelets to extract multiscale information based on an earlier construction for Euclidean data (Mallat, 2012) (see also Mallat (2010); Oyallon et al. (2017); Andreux et al. (2018); Wiatowski & Bölcskei (2015; 2018); Eickenberg et al. (2018)).

*Diffusion wavelets* (Coifman & Maggioni, 2006) are constructed via the diffusion operator

$$\mathbf{P} = \frac{1}{2}(\mathbf{I} + \mathbf{D}^{-1}\mathbf{A}), \tag{1}$$

where $\mathbf{D} \in \mathbb{R}^{n \times n}$ is the diagonal degree matrix.[1] Letting $J$ be a positive integer, we define diffusion wavelets $\mathcal{W}_J = \{\mathbf{\Psi}_j\}_{j=0}^J \cup \{\mathbf{\Phi}_J\}$ by

$$\mathbf{\Psi}_j = \mathbf{P}^{2^{j-1}} - \mathbf{P}^{2^j} = \mathbf{P}^{2^{j-1}}(\mathbf{I} - \mathbf{P}^{2^{j-1}}) \tag{2}$$

for $1 \leq j \leq J$, and $\mathbf{\Psi}_0 = \mathbf{I} - \mathbf{P}$ and $\mathbf{\Phi}_J = \mathbf{P}^{2^J}$. To understand these wavelets, observe that for a given signal $\mathbf{x}$, $\mathbf{\Psi}_1 \mathbf{x}[i] = (\mathbf{P} - \mathbf{P}^2)\mathbf{x}[i]$, the $i$-th entry of $\mathbf{\Psi}_1 \mathbf{x}$, can be interpreted as describing the differences in the behavior of a signal $\mathbf{x}$ with a one-hop neighborhood of $v_i$ to the behavior of $\mathbf{x}$ in a two-hop neighborhood. Similarly, $\mathbf{\Psi}_0 \mathbf{x} = (\mathbf{I} - \mathbf{P})\mathbf{x}$ describes how the behavior of $\mathbf{x}$ at each vertex differs from at its immediate neighbors. Collectively, the filter bank $\mathcal{W}_J = \{\mathbf{\Psi}_j\}_{j=0}^J \cup \{\mathbf{\Phi}_J\}$ acts as a multi-scale feature extractor where $\mathbf{\Phi}_J$ extracts global information from $\mathbf{x}$ (at scale $2^J$) and each $\mathbf{\Psi}_j$ tracks changes across different scales. From the GSP perspective $\mathbf{\Phi}_J$, is interpreted as a low-pass filter and the $\mathbf{\Psi}_j$ are interpreted as band-pass filters that highlight different frequency bands.

Given the bank of diffusion wavelets $\mathcal{W}_J$, the geometric scattering transform is a non-linear multi-layer feature extractor. It defines *scattering coefficients* via alternating sequences of linear maps (chosen to be wavelets) and entry-wise activations, analogous to a neural network. Formally, first- and second-order scattering coefficients are defined by

$$\mathcal{U}[j]\mathbf{x}(v) = \sigma(\mathbf{\Psi}_j \mathbf{x}(v)), \quad \text{and} \tag{3}$$

$$\mathcal{U}[j, j']\mathbf{x}(v) = \mathcal{U}[j']\mathcal{U}[j]\mathbf{x}(v) = \sigma(\mathbf{\Psi}_j \sigma(\mathbf{\Psi}_j \mathbf{x}(v))), \tag{4}$$

for $0 \leq j \leq j' \leq J$, where $\sigma$ is again an activation function. Further, if desired, $m$-th order scattering coefficients for $m \geq 3$ can be defined similarly by $\mathcal{U}[j_1, j_2, \ldots, j_m]\mathbf{x}(v) = \mathcal{U}[j_m]\mathcal{U}[j_1, \ldots, j_{m-1}]\mathbf{x}(v)$.

When used for node-level tasks, one can view the collection of all $\mathcal{U}[j]\mathbf{x}$ and $\mathcal{U}[j, j']\mathbf{x}$ as a new set of node-features that can then be fed into a prediction network. For graph-level tasks, one typically first performs a global aggregation by computing moments $\mathcal{S}[j, q]\mathbf{x} = \sum_{i=1}^n |\mathcal{U}[j]\mathbf{x}(v_i)|^q$, and $\mathcal{S}[j, j', q]\mathbf{x} = \sum_{i=1}^n |\mathcal{U}[j, j']\mathbf{x}(v_i)|^q$ before applying a final prediction head.

Additionally, we note various works have incorporated diffusion wavelets and scattering transforms into high-performing GNNs. For instance Wenkel et al. (2024) and Johnson et al. (2025a) use learnable combinations of the node features and the wavelets filters in each layer, whereas Tong et al. (2024) replaces with dyadic wavelets with wavelets of the form $\mathbf{P}^{t_j} - \mathbf{P}^{t_{j+1}}$, where the $t_j$ are scales that are learned through a differentiable selector matrix. We

---

[1] Various versions of the geometric scattering transform use different normalizations of the diffusion matrix $\mathbf{P}$. All of our theory may be readily adapted to other normalizations following the analysis provided in Perlmutter et al. (2023).

further note that diffusion wavelets and other graph wavelets can be used outside of the scattering transform for exploratory data analysis and other tasks; and that they can be extended to generalized graphs such as directed graphs, hypergraphs, and simplicial complexes (Venkat et al., 2024a;b; Cloninger et al., 2021; Irion & Saito, 2016; Saito et al., 2024; Sun et al., 2025; Viswanath et al., 2026).

## 2.2. The manifold hypothesis

In this section, we consider high-dimensional point clouds $\{v_i\}_{i=1}^n \subseteq \mathbb{R}^D$ and review techniques that aim to find and utilize a latent low-dimensional structure by constructing a graph that may be interpreted as a discrete approximation of a latent underlying data manifold. High-dimensional point clouds arise in many applications, with prominent examples including single-cell data analysis (Ahlmann-Eltze & Huber, 2025; Kröger et al., 2024) and neural data (Kaufman et al., 2016).

Unfortunately, traditional methods often suffer from the curse of dimensionality when $D$ is large. However, it is often possible to overcome these difficulties by uncovering hidden geometric structure, thus reducing the effective dimension of the data. For instance, although neural population activity is recorded in a high-dimensional space whose dimensions correspond to individual neurons, the observed activity often lies on a low-dimensional manifold reflecting the network's behaviorally realizable dynamics (Perich et al., 2025; Sadtler et al., 2014). Such settings lead us to the *manifold hypothesis*: the assumption that the data lies on an unknown (compact, Riemannian) manifold $\mathcal{M}$, i.e., a low-dimensional subset of $\mathbb{R}^D$, whose intrinsic dimension $d$ is much lower than the ambient dimension $D$.

A commonly used approach for data that satisfies the manifold hypothesis is to construct a graph $G = (V, E)$, whose vertices are the data points $\{v_i\}_{i=1}^n$, interpreted as a discretization of the underlying manifold. For instance, the diffusion maps algorithm (Coifman & Lafon, 2006) interprets the diffusion matrix $\mathbf{P}$ as a discrete approximation of an underlying manifold heat kernel. It uses the first $m$ eigenvectors and eigenvalues of $\mathbf{P}$, $\{\mathbf{u}_i\}_{i=1}^m$ and $\{\omega_i\}_{i=1}^m$ to map the data points $v_i$ to $\mathbb{R}^m$ via $v_i \to (\omega_1^t \mathbf{u}_1[i], \ldots, \omega_m^t \mathbf{u}_m[i])$, where $t$ is a hyper-parameter known as diffusion time. Laplacian eigenmaps (Belkin & Niyogi, 2003) proceed similarly, embedding points into a lower-dimensional space using coordinates derived from the low-frequency eigenvectors of the graph Laplacian.

To connect manifold learning to geometric deep learning, we note that a natural way to construct neural networks on a compact Riemannian manifold is to define convolutions using the spectral decomposition of the Laplace-Beltrami operator $\mathcal{L}$. This parallels popular spectral graph neural networks such as Bruna et al. (2014) and Defferrard et al.

(2016), which utilize the eigendecomposition of the graph Laplacian. Papers such as Perlmutter et al. (2020b); Wang et al. (2024), and Wang et al. (2022a) analyze the theoretical stability of such networks, and Chew et al. (2023); Wang et al. (2022b) introduce numerical methods for implementing such networks on point clouds sampled from an unknown manifold, with provable statistical consistency guarantees. Most closely related to our proposed method are Perlmutter et al. (2020b); Chew et al. (2024); Johnson et al. (2025a), which introduce networks based on diffusion wavelets (see Section 2.1).

This work extends diffusion wavelets to the case of vector-valued signals $\mathbf{w} : V \to \mathbb{R}^D$. Where appropriate, we will assume that these vector-valued signals take values lying in the tangent bundle $\mathcal{TM}$ of an unknown manifold, with each $\mathbf{w}(v_i)$ taking values in $\mathcal{T}_{v_i}\mathcal{M}$, the tangent space centered at $v_i$. In this setting, we consider the Connection Laplacian $\Delta$, which is the natural, higher-order analog of the Laplace-Beltrami Operator $\mathcal{L} = -\text{div} \circ \nabla$. (In particular, $\Delta$ is a differential operator which acts upon functions defined on $\mathcal{TM}$ whereas $\mathcal{L}$ acts on functions defined on $\mathcal{M}$.) We note that Battiloro et al. (2024) use this operator and the associated heat semigroup $e^{-t\Delta}$ to define spectral neural networks for tangent-bundle valued data, and also analyze the convergence of such networks as $n \to \infty$ (as discussed in Appendix E).

Our method, detailed in Section 3, extends diffusion wavelets and associated networks to this setting. In short, we define wavelets in terms of a vector diffusion matrix $\mathbf{Q}$, based on Singer & Wu (2012), which introduces a similar operator in the context of constructing vector diffusion maps. Similar to works such as Chew et al. (2022); Johnson et al. (2025a) — which view powers of the standard diffusion operator $\mathbf{P}^t$ as a computationally efficient proxy for the heat semigroup $e^{-t\mathcal{L}}$ associated to the Laplace-Beltrami operator — we view powers of $\mathbf{Q}$ as a proxy for $e^{-t\Delta}$, the heat semigroup associated to the connection Laplacian. This approximation allows us to implement the associated vector diffusion wavelets using sparse multiplications and avoid explicit eigendecompositions, thereby significantly increasing the computational efficiency and scalability our method.

## 2.3. Equivariant GNNs

In this section, we consider geometric graphs where the vertices $\{v_i\}_{i=1}^n \subseteq \mathbb{R}^D$ are not necessarily assumed to lie upon a $d$-dimensional manifold, $d < D$. In this case, however, we can still design a network which aims to utilize intrinsic structure and symmetries of the data.

That is, taking inspiration from Equivariant GNNs (Satorras et al., 2021; Batzner et al., 2022; Han et al., 2022; Duval et al., 2024), we show that our network has desirable symmetries with respect to rotations in Section 3. Specifically,

we let $\mathbf{R} \in \mathbb{R}^{D \times D}$ denote a rotation matrix that we assume acts on our vertex set $\{v_i\}_{i=1}^n \to \{\mathbf{R}v_i\}_{i=1}^n$. We let $\mathbf{x} = \mathbf{x}^{(0)}$ and $\mathbf{w} = \mathbf{w}^{(0)}$ denote initial scalar-valued and vector-valued signals, and we let $\mathbf{x}^{(\ell)}$ and $\mathbf{w}^{(\ell)}$ denote our representations after $\ell$ layers.

The manner in which our network processes the scalar-valued signals is not affected by the action of the rotation $\mathbf{R}$, i.e., we have that $\overline{\mathbf{x}}^{(\ell)} = \mathbf{x}^{(\ell)}$, where $\overline{\mathbf{x}}^{(\ell)}$ is analog of $\mathbf{x}^{(\ell)}$ but for the rotated system. This property is referred to as *rotational invariance*. For the vector-valued signals, we show that rotating the system rotates our representation in the corresponding manner, i.e., $\overline{\mathbf{w}}^{(\ell)} = \mathbf{R}\mathbf{w}^{(\ell)}$. This property is referred to as *rotational equivariance*. Prior work on equivariant GNNs (Satorras et al., 2021; Batzner et al., 2022; Han et al., 2022; Duval et al., 2024) demonstrates that these symmetries provide an effective inductive bias for learning.

## 3. Vector Diffusion Wavelets and VDW-GNNs

As in Section 2, we assume that we are given $F_{\text{scalar}}$ scalar-valued signals $\mathbf{x}_i : V \to \mathbb{R}$, $1 \le i \le F_{\text{scalar}}$, and $F_{\text{vec}}$ vector-valued signals $\mathbf{w}_i : V \to \mathbb{R}^D$, $1 \le i \le F_{\text{vec}}$. As in Section 2.1, when convenient, we will identify signals with vectors. We let $\mathbf{w}_i[j] = \mathbf{w}_i(v_j) \in \mathbb{R}^D$ denote the value of the signal $\mathbf{w}_i$ at each vertex, and when we view $\mathbf{w}_i$ as a vector in $\mathbb{R}^{nD}$, we write $\mathbf{w}_i = [\mathbf{w}_i[1]^\top, \ldots, \mathbf{w}_i[n]^T]^\top = [\mathbf{w}_i[1][1], \ldots, \mathbf{w}_i[1][D], \ldots, \mathbf{w}_i[n][1], \ldots, \mathbf{w}_i[n][D]]^\top$.

To process the vector-valued features, we introduce a new form of diffusion wavelets, inspired by vector diffusion maps (Singer & Wu, 2012) and defined in term of a vector diffusion matrix $\mathbf{Q} \in \mathbb{R}^{nD \times nD}$. For each node $v_i \in \mathbb{R}^D$, we construct a local basis $\{\mathbf{u}_{i,1}, \ldots, \mathbf{u}_{i,D}\}$ which provides a local coordinate system for each node relative to its neighbors. To build this local basis, we let $\mathcal{N}_{v_i} = \{v_j \in V : \{v_i, v_j\} \in E\}$ denote the one-hop neighborhood of $v_i$ and let $n_i = |\mathcal{N}_{v_i}|$ denote the number of neighbors.[2] We then define a relative distance matrix $\mathbf{C}_i \in \mathbb{R}^{D \times n_i}$, whose columns are denoted by $(v_{i_j} - v_i)$ where $v_{i_j}$ is the $j$-th neighbor of $v_i$. In order to give more weight to nearer neighbors, we then rescale $\mathbf{C}_i$ by defining $\mathbf{B}_i = \mathbf{C}_i\mathbf{D}_i$, where $\mathbf{D}_i$ is a diagonal matrix defined by $\mathbf{D}_i[j,j] = \sqrt{K_\epsilon(v_i, v_{i_j})}$, and $K_\epsilon(\cdot, \cdot)$ is a Gaussian kernel with scale $\epsilon$, $K(v_i, v_{i_j}) = \exp(-\|v_i - v_{i_j}\|_2^2/\epsilon)$.

We next compute the singular value decomposition (SVD), $\mathbf{B}_i = \mathbf{U}_i\mathbf{\Sigma}_i\mathbf{V}_i^\top$. We note that, by the definition of the SVD, the matrix $\mathbf{U}_i$ is unitary, and so its columns $\{\mathbf{u}_{i,1}, \ldots, \mathbf{u}_{i,D}\}$ form an orthonormal basis for $\mathbb{R}^D$, which we interpret as a defining a local coordinate system cen-

---

[2] We assume that we have $n_i \ge D$. Otherwise, we add edges between each $v_i$ and its nearest neighbors until $\deg(v_i) \ge D$. For further discussion, see Appendix E.

tered around node $v_i$. We assume throughout that each of the singular values, i.e., the diagonal entries of $\mathbf{\Sigma}_i$, are in decreasing order and that none of the singular values have multiplicity greater than one. This ensures that the SVD is unique up to sign flips of the singular vectors.

For all $i$ and $j$, we define $\mathcal{O}_{i,j} = \mathbf{U}_i\mathbf{U}_j^\top \in \mathbb{R}^{D \times D}$ which *shifts between the local coordinate systems* centered at $v_i$ and $v_j$ as shown in Figure 1. However, since the SVD is only unique up to sign-flips, one may obtain a different SVD, $\mathbf{B}_i' = \mathbf{U}_i'\mathbf{\Sigma}_i(\mathbf{V}')_i^\top$ by replacing both $\mathbf{u}_{i,k}$ and $\mathbf{v}_{i,k}$ with $\mathbf{u}_{i,k}' = -\mathbf{u}_{i,k}$ and $\mathbf{v}_{i,k}' = -\mathbf{v}_{i,k}$ for any fixed $k$. Therefore, in order to ensure that the matrices $\mathcal{O}_{i,j}$ are well-defined (i.e., independent of these sign choices), we employ a *sign-flipping* technique that, when needed, replaces $\mathbf{u}_{i,k}$ with $-\mathbf{u}_{i,k}$. This ensures that $\langle \mathbf{u}_{i,k}, \mathbf{u}_{j,\ell} \rangle$ is always non-negative (full details are provided in Appendix C).

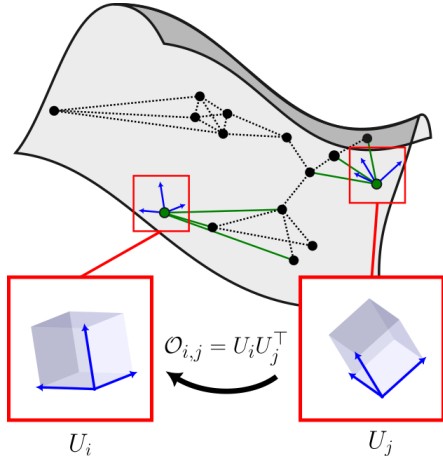

*Figure 1.* An example point cloud of data lying on a continuous manifold. Dashed lines: edges of a $k$-NN graph. Solid green lines: neighbor edges used to generate the local PCA for each node. Blue arrows: local coordinate systems/frames $U_i$ and $U_j$ induced by the local graph structure. $\mathcal{O}_{i,j}$ is the rotation that transforms $U_j$ to $U_i$.

Using these $\mathcal{O}_{i,j}$, we define a $nD \times nD$ vector-valued diffusion matrix in block form by

$$\mathbf{Q}[i,j] = \mathbf{P}[i,j]\mathcal{O}_{i,j} \in \mathbb{R}^{D \times D}.$$

We observe that for $i \ne j$, we have $\mathbf{Q}[i,j] = \mathbf{0}$ unless $\{v_i, v_j\} \in E$ in which case we do not need to compute $\mathcal{O}_{i,j}$. This significantly reduces our computational cost. (Details on computational complexity are given in Appendix D.)

Given $\mathbf{Q}$, we may then define vector diffusion wavelets and scattering coefficients analogous to (2), (3), and (4). Specifically, we define $\widetilde{\mathcal{W}}_J = \{\widetilde{\mathbf{\Psi}}_j\}_{j=0}^J \cup \{\widetilde{\mathbf{\Phi}}_J\}$ by

$$\widetilde{\mathbf{\Psi}}_j = \mathbf{Q}^{2^{j-1}} - \mathbf{Q}^{2^j} = \mathbf{Q}^{2^{j-1}}(\mathbf{I} - \mathbf{Q}^{2^{j-1}}), \quad (5)$$

for $1 \le j \le J$, $\widetilde{\mathbf{\Psi}}_0 = \mathbf{I} - \mathbf{Q}$, and $\widetilde{\mathbf{\Phi}}_J = \mathbf{Q}^{2^J}$.

We then define the vector-valued scattering coefficients of a vector-valued signal $\mathbf{w} : V \to \mathbb{R}^D$ by

$$\widetilde{\mathcal{U}}[j]\mathbf{w}(v) = \sigma(\widetilde{\mathbf{\Psi}}_j\mathbf{w}(v)), \quad \text{and} \tag{6}$$

$$\widetilde{\mathcal{U}}[j,j']\mathbf{w}(v) = \widetilde{\mathcal{U}}[j']\widetilde{\mathcal{U}}[j]\mathbf{w}(v) = \sigma(\widetilde{\mathbf{\Psi}}_{j'}\sigma(\widetilde{\mathbf{\Psi}}_j\mathbf{w}(v))). \tag{7}$$

When it is convenient to think of the signal $\mathbf{w}$ as a vector in $\mathbb{R}^{nd}$, we will write the first-order coefficients as $\widetilde{\mathcal{U}}[j]\mathbf{w} = \sigma(\widetilde{\mathbf{\Psi}}_j\mathbf{w})$ instead of $\widetilde{\mathcal{U}}[j]\mathbf{w}(v) = \sigma(\widetilde{\mathbf{\Psi}}_j\mathbf{w}(v))$ (and similarly with the higher-order coefficients).

Analogous to, e.g., Min et al. (2022; 2021); Tong et al. (2024); Johnson et al. (2025a); Wenkel et al. (2024), in our experiments we use these vector-diffusion wavelets and scattering transforms as the basis for variety of GNNs, which we refer to as Vector Diffusion Wavelet Graph Neural Networks (VDW-GNNs). Additionally, we note in the scalar-valued signal case, Tong et al. (2024) show that the scattering transform can be extended to include non-dyadic diffusion wavelets of the form $\mathbf{P}^{t_j} - \mathbf{P}^{t_{j+1}}$, where $0 = t_0 < t_1 < t_2 \ldots$ is a generic sequence of increasing integers. Vector diffusion wavelets may be readily extended to non-dyadic scales in the same way.

## 3.1. Theoretical Results

The following result establishes frame bounds for the vector diffusion wavelets, similar to results for the scalar-valued diffusion wavelets (Gama et al., 2018; Perlmutter et al., 2023). It implies that the vector diffusion wavelet transform may be stably inverted and is robust to additive noise.[3]

**Theorem 3.1.** *There exists a universal constant $c > 0$ such that for all $\mathbf{w} \in \mathbb{R}^{nD}$ we have*

$$c\frac{d_{\min}}{d_{\max}}\|\mathbf{w}\|_2^2 \le \left\|\widetilde{\mathcal{W}}_J\mathbf{w}\right\|_2^2 \le \frac{d_{\max}}{d_{\min}}\|\mathbf{w}\|_2^2,$$

*where $d_{\min}$ and $d_{\max}$ denote the minimal and maximal vertex degrees and $\left\|\widetilde{\mathcal{W}}_J\mathbf{w}\right\|_2^2 := \sum_{j=0}^J \left\|\widetilde{\mathbf{\Psi}}_j\mathbf{w}\right\|_2^2 + \left\|\widetilde{\mathbf{\Phi}}_J\mathbf{w}\right\|_2^2$.*

We next show that our vector diffusion wavelets and scattering coefficients are equivariant to the actions of the Special Orthogonal group, $SO(D)$, i.e., the set of all $D \times D$ rotation matrices. We assume that our entire system has been subjected to the same global rotation $\mathbf{R} \in SO(D)$ and use bars to denote objects in the rotated coordinate system. Accordingly, $\overline{v}_i = \mathbf{R}v_i$, where $\overline{v}_i = (\overline{v}_i[1], \overline{v}_i[2], \ldots, \overline{v}_i[D])^\top$ denotes the position of the $i$-th vertex in the rotated system, and $\overline{\mathbf{Q}}$ denotes the vector diffusion matrix constructed from the $\overline{v}_i$. Motivated by examples such as when our vector-valued-node features are, e.g., the coordinates of each vertex, we assume that rotating the entire system rotates the values of vector-valued features, giving $\overline{\mathbf{w}}_f = \mathbf{R} \cdot \mathbf{w}$, $\mathbf{R} \cdot \mathbf{w}_f = [(\mathbf{R}\mathbf{w}_f(v_1)^\top), \ldots, (\mathbf{R}\mathbf{w}_f(v_n))^\top]^\top$. Furthermore,

[3]Proofs of all theorems are provided in Appendix A.

we assume that rotations do not change the connectivity structure of the graph, i.e., $\overline{\mathbf{A}} = \mathbf{A}$. The following theorems establish the equivariance of vector diffusion wavelets and associated scattering coefficients. For a visual illustration of Theorem 3.2, see Figure 4 in the appendix.

**Theorem 3.2** (Wavelet Equivariance)**.** *For any vector-valued node feature $\mathbf{w}$, we have, $\overline{\widetilde{\mathbf{\Phi}}}_J\overline{\mathbf{w}} = \mathbf{R} \cdot \widetilde{\mathbf{\Phi}}_J\mathbf{w}$ and*

$$\overline{\widetilde{\mathbf{\Psi}}}_j\overline{\mathbf{w}} = \overline{\widetilde{\mathbf{\Psi}}}_j(\mathbf{R} \cdot \mathbf{w}) = \mathbf{R} \cdot \widetilde{\mathbf{\Psi}}_j\mathbf{w}, \quad \textit{for all} \quad 0 \le j \le J.$$

**Theorem 3.3** (Scattering Equivariance)**.** *Assume that $\sigma$ commutes with rotations, i.e., that $\sigma(\mathbf{R} \cdot \mathbf{w}) = \mathbf{R} \cdot \sigma(\mathbf{w})$ for all rotation matrices $\mathbf{R}$. Then, for all $m \ge 1$, we have*

$$\overline{\widetilde{\mathcal{U}}}[j_1, \ldots, j_m]\overline{\mathbf{w}} = \overline{\widetilde{\mathcal{U}}}[j_1, j_2, \ldots, j_m](\mathbf{R} \cdot \mathbf{w})$$
$$= \mathbf{R} \cdot \widetilde{\mathcal{U}}[j_1, \ldots, j_m]\mathbf{w}.$$

Note that Theorem 3.3 introduces the assumption that $\sigma$ commutes with rotations, which we discuss further in Appendix A.3. Lastly, since relative distance matrices $\mathbf{C}_i$ are defined in terms of relative differences $v_{i_j} - v_i$, this ensures that our method is automatically translation invariant.

# 4. Experimental Results

We evaluate VDW-GNNs across three progressively complex experimental settings: synthetic point clouds, real-world wind velocity measurements, and neural activity data. Our experiments include both node-level and graph-level targets; scalar-valued and vector-valued targets; and inductive classification, regression, and representation learning tasks. Collectively, these experiments demonstrate that VDWs provide a flexible framework for vector-valued learning on geometric data. A detailed discussion of their computational complexity can be found in Appendix D.[4]

## 4.1. Synthetic ellipsoids

We first consider node-level and graph-level tasks on synthetic 3D point clouds randomly sampled from the surface of randomly generated ellipsoids. For this task, we use a VDW-GNN model, which features vector- and scalar-track scattering modules with equivariant hidden feature mixing layers as shown in Figure 2. As baselines, we use (1) VDW-GNN (non-equivariant), an ablated version of our model which treats each vector-valued signal $\mathbf{w} : V \to \mathbb{R}^3$ as three separate scalar-valued signals; (2) LEGS (Tong et al., 2020), a non-equivariant scattering based GNN; two equivariant GNNs, (3) EGNN (Satorras et al., 2021) and (4) TFN (Thomas et al., 2018); and several standard message passing networks: (5) GCN (Kipf & Welling, 2016), (6) GAT

[4]Our code is available at https://github.com/dj408/vdw-gnns

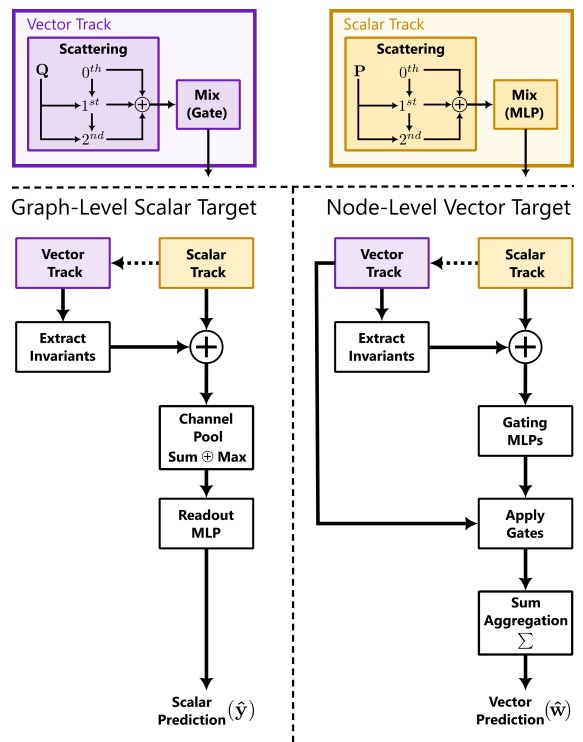

*Figure 2.* Model architecture variants for the ellipsoids experiments. Dotted arrows indicate that the scalar track's hidden features can be used as input in the vector track's mixing layer gate. Individual module descriptions can be found in Appendix F.1.

(Veličković et al., 2018), and (7) GIN (Xu et al., 2019). For a detailed description of our architecture and experimental setup, see Appendix F.

For node features, we use the 3D coordinates of the points. Collectively, our experiments demonstrate the importance of rotational equivariance, with our VDW-GNN achieving comparable performance to other equivariant GNNs with significantly fewer trainable parameters.

*We first consider a graph-level task,* on a data set of $N_G = 512$ graphs, where the goal is to predict the Euclidean diameter of each graph, $\mathrm{diam}(G_j) = \max_{v,v' \in V(G_j)} \|v - v'\|_2$. To construct these graphs, we define random ellipsoids $\mathcal{E}_j = \{(x, y, z) \in \mathbb{R}^3 : x^2/a_j^2 + y^2/b_j^2 + z^2/c_j^2 = 1\}$, $1 \leq j \leq N_G$, where $a_j$, $b_j$, and $c_j$ are i.i.d. randomly generated coefficients with each $a_j \sim \mathcal{N}(3, 0.5)$ and $b_j, c_j \sim \mathcal{N}(1, 0.2)$, and, for convenience, we use $x, y$ and $z$ to denote the coordinates of a point in 3D space, i.e., $v = (x, y, z)$. We note that by construction, we usually have $a_j^2 > b_j^2, c_j^2$ so these ellipsoids are typically more elongated along the $x$-axis than along the $y$- or $z$-axis.

From each ellipsoid $\mathcal{E}_j$, we then sample $n = 128$ points, $\{v_i^{(j)}\}_{i=1}^n$, and construct a $k$-NN graph with $k = 5$. To demonstrate the utility of equivariant models, we rotate each ellipsoid in the test set (and only in the test set) by 90 degrees so that they are most elongated along the $y$-axis.

Our results are shown in Table 1. We observe that while the non-equivariant models are able to perform well on the (non-rotated) validation set, they fail spectacularly on the test set. For example, the non-equivariant version of VDW-GNN has an average validation mean square error (MSE) of 0.3126, but average test MSE of 2.9181. By contrast, the equivariant version of VDW-GNN, as well as the other equivariant networks, do not suffer from this limitation and achieve similar performance on the validation and test sets. Additionally, VDW-GNN outperforms the equivariant baselines with only 59, 779 parameters compared to the 284, 036 in EGNN and the 29, 672, 961 in TFN.

*We next consider a node-level task* where the goal is to learn the value of a function $\mathbf{h} : \mathcal{E} \to \mathbb{R}^3$ defined on the underlying ellipsoid. At each vertex $v_i$, the direction of $\mathbf{h}(v_i)$ is chosen to be the outward normal vector of $\mathcal{E}$. The magnitude is constructed via a randomly generated bandlimited function, generated using the first $K = 16$ eigenfunctions of the Laplace-Beltrami operator. (See Appendix F.4 for details.) Our results, shown in Table 2, generally tell a similar story to the graph-level results in Table 1: the non-equivariant methods perform significantly worse on the rotated test set versus the non-rotated validation set (although the drop in performance is not quite as extreme as before). The performance of VDW-GNN is comparable to the other equivariant GNNs, slightly better than EGNN, and slightly worse than TFN. However, VDW-GNN uses only 57, 250 parameters in comparison to 834, 824 for EGNN and 29, 664, 384 for TFN. The observation that our model far surpasses others in diameter estimation task, but obtains only competitive results on the node-level regression task, may indicate that VDWs are particularly useful for graph-level tasks where it is important to capture global structure, due to the inherent multiscale nature of the wavelet filters. Overall, our results indicate that VDW-GNN is an efficient, lightweight alternative to standard equivariant GNNs.

### 4.2. Wind field reconstruction

We next consider a real-world wind data set previously considered by Battiloro et al. (2024), who introduce a tangent-bundle neural network defined in terms of the connection Laplacian. This data set consists of the earth's mean 10-meter (from the surface) wind field measurements for a single day (January 1, 2016), retrieved from the NOAA Physical Sciences Laboratory (2025). Each sample in the data set has zonal (latitude-aligned) and meridional (longitude-aligned) vector components, as well as the corresponding latitude/longitude coordinates for the location of the measurement. Similar to Battiloro et al. (2024), in our experiment, we aim to reconstruct masked wind velocity measurements. Practically, this task can interpreted as recovering lost signals in a sensor network degradation scenario (e.g., random remote wind sensor failures across the globe).

*Table 1.* Results on the diameter prediction task (mean $\pm$ std. across 5-fold CV). Validation MSE is the (average) of the best score achieved during training for each fold. Best is **bolded**; second-best is underlined. VDW-GNN outperforms other equivariant methods with a fraction of the parameter count.

| Model | Validation MSE $\downarrow$ | (Rotated) test MSE $\downarrow$ | Parameter count |
|---|---|---|---|
| VDW-GNN (Ours) | $0.0035 \pm 0.0011$ | $\mathbf{0.0037 \pm 0.0022}$ | $59,779$ |
| VDW-GNN (non-equivariant) (Ours) | $0.3126 \pm 0.5386$ | $2.9181 \pm 5.6549$ | $14,481$ |
| LEGS | $1.1812 \pm 2.5469$ | $2.2759 \pm 3.0514$ | $19,525$ |
| GCN | $0.1684 \pm 0.3247$ | $1.4566 \pm 2.8307$ | $60,801$ |
| GAT | $0.3964 \pm 0.6675$ | $1.4443 \pm 2.3166$ | $61,313$ |
| GIN | $0.4778 \pm 0.6908$ | $1.6425 \pm 2.5529$ | $93,825$ |
| EGNN (2-layer) | $0.0326 \pm 0.0310$ | $0.0284 \pm 0.0249$ | $284,036$ |
| TFN (4-layer) | $0.0787 \pm 0.0070$ | $0.0828 \pm 0.0122$ | $29,672,961$ |

*Table 2.* Node-level results (mean $\pm$ std. across five-fold CV). Validation MSE is the (average) of the best score achieved during training for each fold. Best test MSE score is **bolded**; second-best is underlined. VDW-GNN is the second best method to TFN, but with less than 0.2% of the parameters.

| Model | Validation MSE $\downarrow$ | (Rotated) test MSE $\downarrow$ | Parameter count |
|---|---|---|---|
| VDW-GNN (Ours) | $0.1525 \pm 0.0069$ | $0.1513 \pm 0.0080$ | $57,250$ |
| VDW-GNN (non-equivariant) (Ours) | $0.2650 \pm 0.2654$ | $0.3552 \pm 0.2867$ | $14,611$ |
| LEGS | $0.2224 \pm 0.2167$ | $0.3575 \pm 0.3239$ | $15,335$ |
| GCN | $0.3528 \pm 0.3075$ | $0.4820 \pm 0.3669$ | $44,451$ |
| GAT | $0.3594 \pm 0.3257$ | $0.4792 \pm 0.3639$ | $44,963$ |
| GIN | $0.3175 \pm 0.2839$ | $0.4536 \pm 0.3512$ | $77,475$ |
| EGNN (7-layer) | $0.1949 \pm 0.0297$ | $0.1960 \pm 0.0311$ | $834,824$ |
| TFN (4-layer) | $0.1137 \pm 0.0032$ | $\mathbf{0.1145 \pm 0.0039}$ | $29,664,384$ |

We first subsample the data set to 2000 wind measurements. We split those 2000 measurements, 70/30, into observed and masked sets, and then further split the masked set 10/10/10 to obtain masked-training, masked-validation, and masked-test sets. For all points in the masked set, we replace wind vector features with the component-wise mean vector of the observed set, and set a reconstruction target as the original wind vector. Here, we wish to demonstrate the utility of our method for processing features that take values in the tangent plane of the 2D sphere lying in 3D space. Accordingly, we embed measurement locations on the unit sphere by converting latitude/longitude to 3D Earth-centered, Earth-fixed (ECEF) Cartesian coordinates, and normalize by the radius of the Earth (assuming sphericity). We also lift the 2D wind vector features into the same 3D global coordinate system (details in Appendix G).

We then construct a modified $k$-NN graph ($k = 3$) using Euclidean distance in 3D in which (i) observed nodes are connected to their $k$ nearest observed neighbors with symmetric (undirected) edges, and (ii) masked nodes receive only directed incoming edges from their $k$ nearest observed nodes, observed $\rightarrow$ masked, (i.e., we take $\mathbf{D}$ to be the out-degree matrix when computing $\mathbf{P}$). This modification prevents non-informative messages masked nodes from averaging with the observed set. Edge weights are initialized as inverse

distance, $w_{ij} = 1/d_{ij}$, and then lazily-normalized by target node so that each node's incoming weights sum to $0.5$.

We train all models with the mean squared error (MSE) loss function, using the MSE of the masked-validation set to enforce early stopping and prevent overfitting. We evaluate the models both on their MSE on the masked-test set wind measurements. Akin to the ellipsoids experiments in Section 4.1, we also compute their MSE after a random 3D rotation is applied to the test set. In the latter case, we use rejection sampling of randomly generated candidate 3D rotation matrices, until a matrix with a rotation angle between 90 and 160 degrees is found.

We again test LEGS, GCN, GAT, GIN, EGNN, and TFN as baselines, as well as the DD-TNN ('Domain-Discretized Tangent Bundle Neural Network') model from Battiloro et al. (2024). We note that somewhat similarly to our method, DD-TNN uses local SVD to define matrices $\mathcal{O}_{i,j}$ which are then used to construct an approximation of the connection Laplacian, also referred to as a sheaf Laplacian. DD-TNN then uses this sheaf Laplacian to define a spectral convolutional network. Note that in order to implement EGNN and TFN for this task, we decouple (i) measurement points' global positions, from (ii) these points' wind vectors, such that message passing occurs between 'position

on earth' neighbors, but messages update in 'wind vector space'. We repeat our experiment five times with unique sampling and rotation seeds. More experimental details are found in Appendix G. Aggregated test set results are in Table 3.

We use a simple VDW-GNN model for this task, in which we apply a single layer of vector diffusion wavelets and then, for each node, concatenate the wind vectors and edge weights (inverse distances) of its three closest neighbors. We then feed this transformed representation into a three-layer MLP. (Preliminary experiments indicated that this outperformed more complex models on this data set.) Our model achieves the best test set error and second-best rotated-test set error, with far fewer parameters than its competitors, EGNN and GCN. It substantially outperforms DD-TNN, which also was designed to process tangent-bundle valued features. Our model also achieves the fastest per-epoch training time, with the important caveat that the diffusion operator $\mathbf{Q}$ is constructed once and cached before training, which takes approximately 1.1 seconds.[5]

### 4.3. Multi-channel neural recordings

Our final experiment considers a neural activity data set from Kaufman et al. (2016), processed by Gosztolai (2023), and featured in three recent works that developed representation learning models for neural data: MAnifold Representation Basis LEarning (MARBLE) (Gosztolai et al., 2025), Consistent EmBeddings of high-dimensional Recordings using Auxiliary variables (CEBRA) (Schneider et al., 2023), and Latent Factor Analysis for Dynamical Systems (LFADS) (Pandarinath et al., 2018). More details on these models are in Appendix H.

The data set consists of 24-channel neural activity time-series data, recorded as a trained macaque (a species of monkey native to Asia and North Africa) physically moved their hand on a screen towards one of seven targets ("conditions") in different 2D directions from a central position in order to obtain a reward. It contains data from 44 separate experimental days, where each day has between 189 and 607 trials (mean: 333; total: 14,660). Formally, the $j$-th trial from day $i$ corresponds to a time series of length $T$ taking values in $\mathbb{R}^{24}$, i.e., $\mathbf{x}^{(i,j)} : [T] \to \mathbb{R}^{24}$. The modeling task is to learn low-dimensional embeddings of the per-timepoint neural signals among trials, within days.[6] Embedding quality is assessed by the accuracy of a simple classifier taking them as input and predicting the corresponding condition.

We use the version of the data set provided at Gosztolai

(2023), and we generally follow the same preprocessing methods as their corresponding paper (Gosztolai et al., 2025), except we do not apply PCA to the processed 24-dimensional features (for any model). Our setup also differs from previous treatments of this data in that we conduct our experiments in an inductive (rather than transductive) setting. This represents a more challenging task, and ensures that model embeddings reflect a generalization of the representation function to unseen trials, rather than memorization of the training graph.

As in Gosztolai et al. (2025), we construct a continuous $k$-nearest neighbors (CkNN) graph (Berry & Sauer, 2019), whose nodes are per-timepoint samples in a multi-trial, within-day neural state space and the edge set is constructed to have a density-adjusted number of neighbors. To construct vector-valued node features, we again follow the lead of Gosztolai et al. (2025) by numerically differentiating the neural activity at each node to obtain 24-dimensional "neural velocities." Our model applies vector diffusion wavelets to these velocities, concatenates wavelets' outputs, feeds them into a four-layer MLP, and trains with a supervised contrastive loss (Khosla et al., 2020) function enhanced with a custom, hard-negatives-focused sampling module.

Performance is assessed using the same probing methodology in Gosztolai et al. (2025): after learning a three-dimensional embedding of timepoints (the most challenging setting used in Gosztolai et al. (2025)'s experiments), we flatten and concatenate timepoint embeddings per trial, train a support vector machine (SVM) on these aggregated embeddings, and evaluate classification accuracy on held-out trial aggregated embeddings for each day. We report aggregate results across all 44 days (performing five-fold cross-validation on separate models per day). As shown in Table 4, our VDW-GNN model achieves an overall mean classification accuracy of 66%, slightly out-performing MARBLE, which achieves 65% mean accuracy. However, across experimental days, these models' results distributions are not significantly different under a two-sided Wilcoxon test (see Figure 5, in Appendix H). Nonetheless, our model both trains much faster and uses far fewer parameters than MARBLE, and significantly outperforms CEBRA and LFADS. For further details on the data set, our experimental setup, and each of the models, see Appendix H.

## 5. Limitations

The main limitations of our method are:

1. Because VDWs depend on local PCA estimates of tangent space bases, their reliability may degrade when neighborhoods are too small or sparse to support accurate estimation of local geometry. We discuss mitigation techniques for such settings in Appendix E.

---

[5] On our hardware, discussed in Appendix D. Note that our model's per-epoch time cost could be reduced even further in the single-layer case by precomputing the wavelet transform step.

[6] The inconsistency of exact neural probe placement between experimental days prevents simple pooling of trials across days.

*Table 3.* Wind field reconstruction task results on the masked-test set, (mean ± std.) across five repetitions. Best test MSE score is **bolded**; second-best is underlined. VDW-GNN again achieves best or second-best results among comparable models with far fewer parameters.

| Model | MSE ↓ | Rotated MSE ↓ | Sec. per epoch | Best epoch | Parameter count |
|---|---|---|---|---|---|
| VDW-GNN (ours) | **2.7546 ± 0.3671** | 3.2266 ± 0.5255 | **0.0041 ± 0.0007** | 93 ± 41 | 20,099 |
| LEGS | 11.2551 ± 2.3892 | 27.5993 ± 9.2066 | 0.0080 ± 0.0003 | 258 ± 132 | 21,319 |
| GCN | 2.8662 ± 0.4264 | 3.5242 ± 0.6116 | 0.0043 ± 0.0012 | 41 ± 17 | 50,435 |
| GAT | 3.2784 ± 0.2189 | 3.8239 ± 0.1370 | 0.0053 ± 0.0039 | 54 ± 45 | 34,179 |
| GIN | 3.1754 ± 0.1940 | 3.5410 ± 0.3372 | 0.0045 ± 0.0030 | 20 ± 4 | 50,435 |
| DD-TNN | 12.7592 ± 0.9288 | 14.8010 ± 1.8416 | 0.0114 ± 0.0003 | 118 ± 110 | 17,374 |
| EGNN (1-layer) | 2.7651 ± 0.3089 | **2.7651 ± 0.3089** | 0.0120 ± 0.0166 | 448 ± 406 | 134,402 |
| TFN (2-layer) | 3.9898 ± 0.1895 | 3.9898 ± 0.1895 | 0.0128 ± 0.0034 | 88 ± 65 | 729,184 |

*Table 4.* Summary results for the neural data experiment. Classifier accuracy: the mean (across 44 days) of mean SVM classifier accuracies (across five-fold CV within days), ± standard deviation (of daily CV means). Best is **bolded**; second-best is underlined.

| Model | Classifier accuracy ↑ | Sec. per epoch | Best epoch | Parameter count |
|---|---|---|---|---|
| VDW-GNN | **0.66 ± 0.14** | 0.23 ± 0.05 | 73 ± 172 | 164,868 |
| MARBLE | 0.65 ± 0.12 | 1.41 ± 0.63 | 20 ± 57 | 1,443,004 |
| CEBRA | 0.57 ± 0.09 | 0.40 ± 0.04 | 685 ± 3040 | 11,171 |
| LFADS | 0.35 ± 0.08 | 0.44 ± 0.11 | 5 ± 34 | 1,068,800 |

2. While VDWs are typically highly parameter efficient, they may be sensitive to key hyperparameters with certain data sets. For instance, tuning choice of diffusion scales employed to construct a filter bank of VDWs, aiming to capture key signal bands in the data, is often a nontrivial but crucial step. Moreover, if the graph structure is not given, the scales may also need to be tuned in tandem with graph-construction parameters such as the value $k$ in a $k$-NN or an the edge-weighting kernel, if applicable.

3. Our method is motivated by the manifold hypothesis, i.e., that belief that data our lies approximately on a smooth, low-dimensional manifold. If this assumption is violated (due to noise or lack of manifold structure, etc.), our method has a less principled derivation and may not perform well.

## 6. Conclusion

We have introduced a novel version of the diffusion wavelets for vector-valued signals, and proved theoretical results demonstrating their frame properties and rotational equivariance. Empirically, we have shown that these wavelets may be incorporated into geometric graph neural networks and effectively applied to synthetic point cloud data, as well as real-world data sets derived from wind field or neural activity measurements.

## Acknowledgments

DJ, MP and SK were funded in part by NSF-DMS-2327211; AS by NSF-DMS-2136090 and NSF-DMS-2408912; MP by NSF-OIA-2242769; DN by NSF-DMS-2408912. SK was also funded in part by NSF Career Grant 2047856.

## Impact Statement

This paper presents work whose goal is to advance the field of Machine Learning. There are many potential societal consequences of our work, none which we feel must be specifically highlighted here.

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

# A. Proofs of Main Theorems

In this section, we will provide the proofs of Theorems 3.1, 3.2, and 3.3. Additionally we recall that, as noted in Section 3, in the scalar-valued signal case, Tong et al. (2024) showed that the scattering transform could be extended to include non-dyadic diffusion wavelets of the form $\mathbf{P}^{t_j} - \mathbf{P}^{t_{j+1}}$ where $0 = t_0 < t_1 < t_2 \ldots$ is a generic sequence of increasing integers and that vector diffusion wavelets may be readily extended to non-dyadic scales. We note that it is straightforward to extend all of our theoretical analysis to wavelets and scattering transforms constructed in this way.

## A.1. The Proof of Theorem 3.1

*Proof.* Let $\mathbf{Q}'$ be the $D \times D$ matrix defined in the same manner as $\mathbf{Q}$, but with the $D \times D$ identity matrix in place of $\mathcal{O}_{i,j}$, i.e.,

$$\mathbf{Q}'[i,j] = \mathbf{P}[i,j]\mathbf{I}.$$

Let $\widetilde{\mathcal{W}}'_J = \{\widetilde{\boldsymbol{\Psi}}'_j\}_{j=0}^J \cup \{\widetilde{\boldsymbol{\Phi}}'_J\}$ be analogous to $\widetilde{\mathcal{W}}_J$, but with $\mathbf{Q}'$ in place of $\mathbf{Q}$, i.e., $\widetilde{\boldsymbol{\Psi}}'_0 = \mathbf{I} - \mathbf{Q}'$,

$$\widetilde{\boldsymbol{\Psi}}'_j = (\mathbf{Q}')^{2^{j-1}} - (\mathbf{Q}')^{2^j}, \quad 1 \le j \le J,$$

and $\widetilde{\boldsymbol{\Phi}}'_J = (\mathbf{Q}')^{2^J}$.

The following lemma shows that a result similar to Theorem 3.1 holds for $\widetilde{\mathcal{W}}'_J$.

**Lemma A.1.** *For all $\mathbf{w} \in \mathbb{R}^{nD}$, we have*

$$c\frac{d_{\min}}{d_{\max}}\|\mathbf{w}\|_2^2 \le \left\|\widetilde{\mathcal{W}}'_J\mathbf{w}\right\|_2^2 := \sum_{j=0}^J \left\|\widetilde{\boldsymbol{\Psi}}'_j\mathbf{w}\right\|_2^2 + \left\|\widetilde{\boldsymbol{\Phi}}'_J\mathbf{w}\right\|_2^2 \le \frac{d_{\max}}{d_{\min}}\|\mathbf{w}\|_2^2,$$

*where $c > 0$ is a universal constant.*

We also need the following lemma which provides a simplified expression for $\mathbf{Q}^m$.

**Lemma A.2.** *For all $m \ge 0$, we may write $\mathbf{Q}^m$ in block form as*

$$\mathbf{Q}^m[i,j] = \mathbf{P}^m[i,j]\mathcal{O}_{i,j},$$

*where $\mathbf{P}^m[i,j] \in \mathbb{R}$ is the $i,j$-th entry of $\mathbf{P}^m$.*

For proofs of Lemma A.1 and A.2, please see Appendices B.1 and B.2.

Now, let $\mathbf{w} = [\mathbf{w}[1]^\top, \ldots, \mathbf{w}[n]^\top]^\top \in \mathbb{R}^{nd}$ be a vector-valued signal (so that $\mathbf{w}[k] = \mathbf{w}(v_k) \in \mathbb{R}^D$). Define $\mathbf{y} \in \mathbb{R}^{nD}$ by $\mathbf{y} = [\mathbf{y}[1]^\top, \ldots, \mathbf{y}[n]^\top]$, where

$$\mathbf{y}[k] = \mathbf{U}_k\mathbf{w}[k].$$

Note that since $\mathbf{U}_k$ is unitary, we have $\|\mathbf{y}[k]\|_2 = \|\mathbf{w}[k]\|_2$ for all $k$, which further implies that $\|\mathbf{y}\|_2 = \|\mathbf{w}\|_2$. Next observe that by Lemma A.2 we have, for $1 \le j \le J$,

$$\widetilde{\boldsymbol{\Psi}}_j[i,k] = \mathbf{Q}^{2^{j-1}}[i,k] - \mathbf{Q}^{2^j}[i,k] = \mathbf{P}^{2^{j-1}}[i,k]\mathcal{O}_{i,k} - \mathbf{P}^{2^j}[i,k]\mathcal{O}_{i,k} = \widetilde{\boldsymbol{\Psi}}'_j[i,k]\mathcal{O}_{i,k},$$

where in the final equality we use the fact that $\widetilde{\boldsymbol{\Psi}}'_j[i,k] = \left(\mathbf{P}^{2^{j-1}}[i,k] - \mathbf{P}^{2^j}[i,k]\right)\mathbf{I}$. Similarly, we have $\widetilde{\boldsymbol{\Psi}}_0[i,k] =$

$\widetilde{\mathbf{\Psi}}_0'[i,k]\mathcal{O}_{i,k}$ and $\widetilde{\mathbf{\Phi}}_J[i,k] = \widetilde{\mathbf{\Phi}}_J'[i,k]\mathcal{O}_{i,k}$. Therefore, for all $0 \le j \le J$, we have

$$
\begin{aligned}
(\widetilde{\mathbf{\Psi}}_j\mathbf{y})[i] &= \sum_{k=1}^n \widetilde{\mathbf{\Psi}}_j[i,k]\mathbf{y}[k] \\
&= \sum_{k=1}^n \widetilde{\mathbf{\Psi}}_j'[i,k]\mathcal{O}_{i,k}\mathbf{U}_k\mathbf{w}[k] \\
&= \sum_{k=1}^n \widetilde{\mathbf{\Psi}}_j'[i,k]\mathbf{U}_i\mathbf{U}_k^\top\mathbf{U}_k\mathbf{w}[k] \\
&= \mathbf{U}_i\sum_{k=1}^n \widetilde{\mathbf{\Psi}}_j'[i,k]\mathbf{w}[k] \\
&= \mathbf{U}_i((\mathbf{\Psi}_j'\mathbf{w})[i]),
\end{aligned}
$$

and likewise $(\widetilde{\mathbf{\Phi}}_J\mathbf{y})[i] = \mathbf{U}_i((\widetilde{\mathbf{\Phi}}_J'\mathbf{w})[i])$. Since $\mathbf{U}_i$ is unitary, this implies that

$$
\sum_{j=0}^J \left\|\widetilde{\mathbf{\Psi}}_j'\mathbf{y}\right\|_2^2 + \left\|\widetilde{\mathbf{\Phi}}_J'\mathbf{y}\right\|_2^2 = \sum_{j=0}^J \left\|\widetilde{\mathbf{\Psi}}_j\mathbf{w}\right\|_2^2 + \left\|\widetilde{\mathbf{\Phi}}_J\mathbf{w}\right\|_2^2. \tag{8}
$$

However, by Lemma A.1, we have

$$
\begin{aligned}
c\,\frac{d_{\min}}{d_{\max}}\|\mathbf{w}\|_2^2 = c\,\frac{d_{\min}}{d_{\max}}\|\mathbf{y}\|_2^2 \\
\le \sum_{j=0}^J \left\|\widetilde{\mathbf{\Psi}}_j'\mathbf{y}\right\|_2^2 + \left\|\widetilde{\mathbf{\Phi}}_J'\mathbf{y}\right\|_2^2 \\
\le \frac{d_{\max}}{d_{\min}}\|\mathbf{y}\|_2^2 \\
= \frac{d_{\max}}{d_{\min}}\|\mathbf{w}\|_2^2.
\end{aligned}
$$

Combining this with (8) completes the proof.

$\square$

## A.2. The Proof of Theorem 3.2

To prove Theorem 3.2, we need the following lemma which establishes the equivariance of the powered vector diffusion matrix $\mathbf{Q}^m$, which is also illustrated by Figure 3. For a proof, please see Appendix B.3.

**Lemma A.3.** *For any $m \ge 0$, and any vector-valued node feature $\mathbf{w} \in \mathbb{R}^{nD}$ we have*

$$
\overline{\mathbf{Q}}^m\overline{\mathbf{w}} = \overline{\mathbf{Q}}^m(\mathbf{R}\cdot\mathbf{w}) = \mathbf{R}\cdot(\mathbf{Q}^m\mathbf{w}).
$$

*The proof of Theorem 3.2.* We first observe that the claim $\overline{\widetilde{\mathbf{\Phi}}}_J\overline{\mathbf{w}} = \mathbf{R}\cdot\widetilde{\mathbf{\Phi}}_j\mathbf{w}$ follows immediately from Lemma A.3, setting $m = 2^J$.

Now, fix $0 \le j \le J$, and observe that we may write $\widetilde{\mathbf{\Psi}}_j = \mathbf{Q}^{t_1} - \mathbf{Q}^{t_2}$ where $t_1 = 0, t_2 = 1$ if $j = 0$ and otherwise we have $t_1 = 2^{j-1}, t_2 = 2^j$. Thus, using Lemma A.3, we observe

$$
\begin{aligned}
\overline{\widetilde{\mathbf{\Psi}}}_j\overline{\mathbf{w}} &= (\overline{\mathbf{Q}}^{t_1} - \overline{\mathbf{Q}}^{t_2})\overline{\mathbf{w}} \\
&= \overline{\mathbf{Q}}^{t_1}\overline{\mathbf{w}} - \overline{\mathbf{Q}}^{t_2}\overline{\mathbf{w}} \\
&= \mathbf{R}\cdot(\mathbf{Q}^{t_1}\mathbf{w}) - \mathbf{R}\cdot(\mathbf{Q}^{t_2}\mathbf{w}) \\
&= \mathbf{R}\cdot((\mathbf{Q}^{t_1} - \mathbf{Q}^{t_2})\mathbf{w}) \\
&= \mathbf{R}\cdot(\widetilde{\mathbf{\Psi}}_j\mathbf{w}),
\end{aligned}
$$

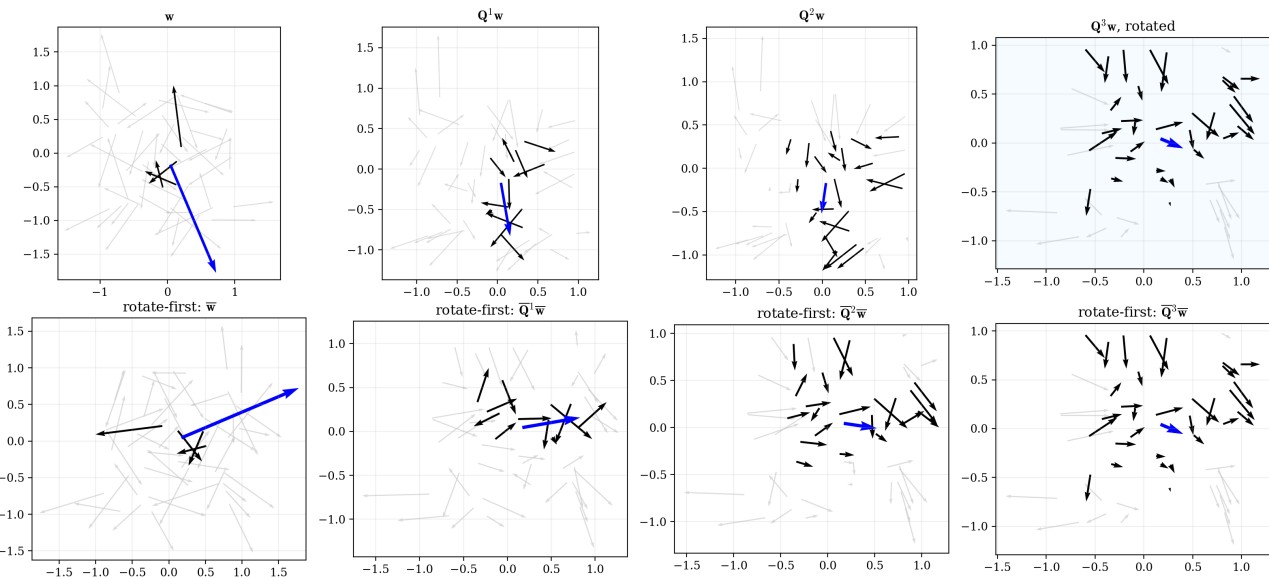

*Figure 3.* Illustration of rotational equivariance of $\mathbf{Q}^m$, applied to a 2D vector field. The vector field was generated with random uniform sampling of points $x_i \sim \mathcal{U}([-1, 1]^2)$, angles $\theta_i \sim \mathcal{U}[0, 2\pi)$, and magnitudes $m_i \sim \mathcal{U}[0.4, 1.0]$ such that $w_i = m_i[\cos\theta_i, \sin\theta_i]$. The central vector (the magnitude of which was made to be the largest for illustrative purposes) is colored blue. The top row shows $\mathbf{Q}^m$ applied to vectors in an unrotated vector field where we first used the sample points $x_i$ to construct a $k$-NN graph, $k = 3$. The bottom row shows the same system, rotated 90 degrees counter-clockwise, then diffused by $\overline{\mathbf{Q}}$. The black vectors highlight those involved in the next diffusion step, specifically, the vectors from which the central vector will receive (indirect) diffusion messages: first, its (symmetric) $k$-nearest neighbor vectors, then neighbors of neighbors, and so on. After three diffusion steps, the top (unrotated) system is rotated 90 degrees counter-clockwise, like the bottom system was initially. This is shown in the top-right panel with a tinted background. Notably, this panel is identical to the panel immediately below it, thereby demonstrating the equivariance of $\mathbf{Q}^m$, since diffusing and then rotating yields the same result as rotating and then diffusing.

where in the final equality we use the fact that $\mathbf{R} \cdot (\mathbf{w}_1 - \mathbf{w}_2) = \mathbf{R} \cdot \mathbf{w}_1 - \mathbf{R} \cdot \mathbf{w}_2$ for any vector-valued node signals $\mathbf{w}_1$ and $\mathbf{w}_2$. $\qquad\square$

### A.3. The Proof of Theorem 3.3

Before proving Theorem 3.3 we briefly discuss the assumption that $\sigma$ commutes with rotations. To understand why this condition is necessary, observe that Theorem 3.2 implies that

$$\overline{\widetilde{\mathcal{U}}}[j_1]\mathbf{w}(v) = \sigma(\overline{\widetilde{\mathbf{\Psi}}}_{j_1}\mathbf{w}(v)) = \sigma(\mathbf{R} \cdot \widetilde{\mathbf{\Psi}}_{j_1}\mathbf{w}(v)).$$

Therefore, Theorem 3.3 will not hold without this assumption. We also note that we may readily construct activations $\sigma$ satisfying this assumption by defining $\sigma(\mathbf{w})(v) = \sigma_{\text{radial}}(\|\mathbf{w}(v)\|_2)\frac{\mathbf{w}(v)}{\|\mathbf{w}(v)\|_2}$ when $\mathbf{w}(v) \neq 0$, and $\sigma(\mathbf{w})(v) = 0$ when $\mathbf{w}(v) = 0$, where $\sigma_{\text{radial}}$ is any real-valued function.

*Proof.* We argue by induction. For the case case, $m = 1$, we use the assumption that $\mathbf{R}$ commutes with rotations to see that

$$\overline{\widetilde{\mathcal{U}}}[j_1]\overline{\mathbf{w}} = \sigma\left(\overline{\widetilde{\mathbf{\Psi}}}_{j_1}\overline{\mathbf{w}}\right) = \sigma\left(R \cdot \widetilde{\mathbf{\Psi}}_{j_1}\mathbf{w}\right) = R \cdot \sigma\left(\widetilde{\mathbf{\Psi}}_{j_1}\mathbf{w}\right) = R \cdot (\widetilde{\mathcal{U}}[j_1]\mathbf{w}).$$

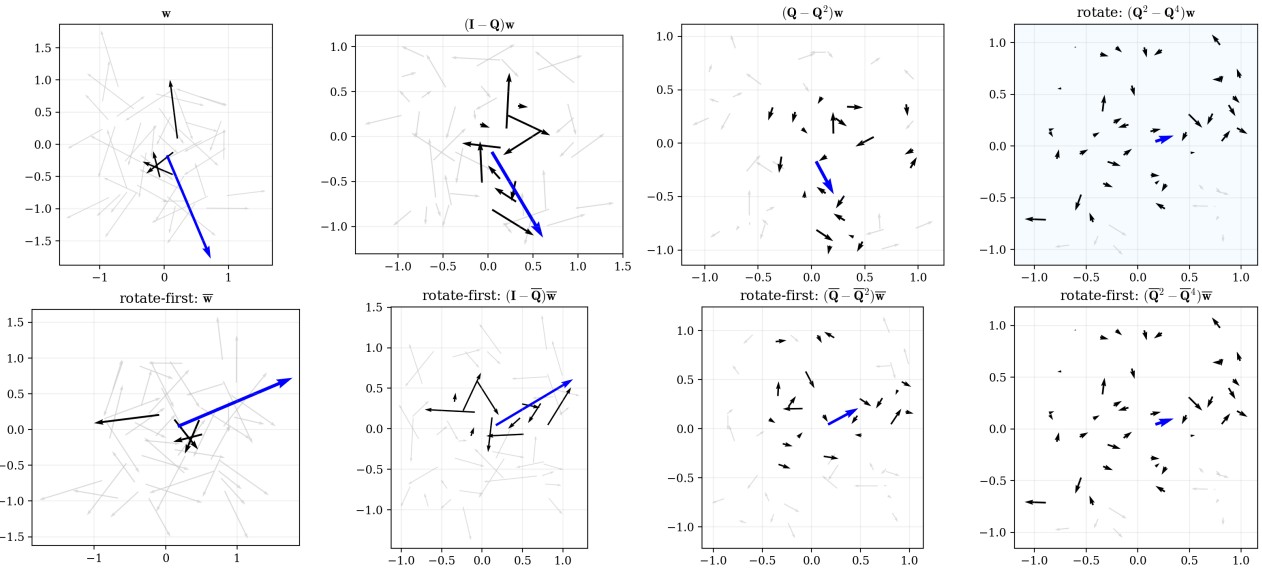

*Figure 4.* Illustration of rotational equivariance of vector diffusion wavelets applied to a vector field (defined as in Figure 3). Here, the top row shows vector diffusion wavelets constructed from $\mathbf{Q}$ applied to vectors in the unrotated vector field; the bottom row shows the same system, rotated 90 degrees counter-clockwise, then diffused using wavelets constructed from $\overline{\mathbf{Q}}$. The black vectors again highlight those from which the central vector will receive (indirect) messages in the next diffusion step. After three wavelet diffusion steps, the top (unrotated) system is rotated 90 degrees counter-clockwise, like the bottom system was initially, as shown in the top-right panel with the tinted background. The fact that the diffused vector fields in the rightmost column are equivalent shows the rotational equivariance of the wavelets.

Now assume the result holds for some $m \geq 1$. Then by the inductive hypothesis, we have

$$\overline{\widetilde{\mathcal{U}}}[j_1, \ldots, j_m, j_{m+1}]\overline{\mathbf{w}} = \overline{\widetilde{\mathcal{U}}}[j_{m+1}]\overline{\widetilde{\mathcal{U}}}[j_1, \ldots, j_m]\overline{\mathbf{w}}$$
$$= \overline{\widetilde{\mathcal{U}}}[j_{m+1}](\mathbf{R} \cdot \widetilde{\mathcal{U}}[j_1, j_2, \ldots, j_m]\mathbf{w})$$
$$= \mathbf{R} \cdot (\overline{\widetilde{\mathcal{U}}}[j_{m+1}]\widetilde{\mathcal{U}}[j_1, j_2, \ldots, j_m]\mathbf{w})$$
$$= \mathbf{R}(\widetilde{\mathcal{U}}[j_1, j_2, \ldots, j_m, j_{m+1}]\mathbf{w}).$$

$\square$

# B. Proofs of auxiliary lemmas

### B.1. The Proof of Lemma A.1

We first recall the following result from Perlmutter et al. (2023) which shows that the diffusion wavelets $\mathcal{W}_J$ are a non-expansive frame on the weighted inner product space defined by $\langle \mathbf{x}_1, \mathbf{x}_2 \rangle_{\mathbf{D}^{1/2}} = \langle \mathbf{D}^{1/2}\mathbf{x}_1, \mathbf{D}^{1/2}\mathbf{x}_2 \rangle_2$, with corresponding norm $\|\mathbf{x}\|_{\mathbf{D}^{1/2}} = \|\mathbf{D}^{1/2}\mathbf{x}\|_2$. (See also Proposition 4.1 of Gama et al. (2018) and Theorem 1 of Tong et al. (2024).) The key to the proof of Proposition B.2 is to combine this result with the inequality (9), stated below, that relates this weighted norm to the unweighted $\ell^2$ norm.

**Proposition B.1** (Proposition 2.2 of Perlmutter et al. (2023)). *$\mathcal{W}_J$ is a nonexpansive frame, i.e., there exists a universal constant $c > 0$, which in particular is independent of $J$, and the geometry of $G$ such that*

$$c\|\mathbf{x}\|_{\mathbf{D}^{1/2}}^2 \leq \|\mathcal{W}_J\mathbf{x}\|_{\mathbf{D}^{1/2}}^2 := \sum_{j=0}^{J} \|\mathbf{\Psi}_j\mathbf{x}\|_{\mathbf{D}^{1/2}}^2 + \|\mathbf{\Phi}_J\mathbf{x}\|_{\mathbf{D}^{1/2}}^2 \leq \|\mathbf{x}\|_{\mathbf{D}^{1/2}}^2 \ \ \text{for all } \mathbf{x} \in \mathbb{R}^n.$$

We use Proposition B.1 to prove the following corollary which analyzes the frame bounds of the diffusion wavelets with respect to the unweighted $\ell^2$ norm.

**Corollary B.2.** *For all* $\mathbf{x} \in \mathbb{R}^n$, *we have*

$$c \frac{d_{\min}}{d_{\max}} \|\mathbf{x}\|_2^2 \leq \|\mathcal{W}_J \mathbf{x}\|_2^2 := \sum_{j=0}^{J} \|\mathbf{\Psi}_j \mathbf{x}\|_2^2 + \|\mathbf{\Phi}_J \mathbf{x}\|_2^2 \leq \frac{d_{\max}}{d_{\min}} \|\mathbf{x}\|_2^2,$$

*where* $d_{\min}$ *and* $d_{\max}$ *denote the minimal and maximal vertex degrees and* $c > 0$ *is a universal constant.*

*Proof.* Let $\mathbf{x} \in \mathbb{R}^n$. It is straightforward to see that

$$d_{\min} \|\mathbf{x}\|_2^2 \leq \|\mathbf{x}\|_{\mathbf{D}^{1/2}}^2 \leq d_{\max} \|\mathbf{x}\|_2^2. \tag{9}$$

Therefore, by Proposition B.1, we have

$$\sum_{j=0}^{J} \|\mathbf{\Psi}_j \mathbf{x}\|_2^2 + \|\mathbf{\Phi}_J \mathbf{x}\|_2^2 \leq d_{\max} \left( \sum_{j=0}^{J} \|\mathbf{\Psi}_j \mathbf{x}\|_{\mathbf{D}^{1/2}}^2 + \|\mathbf{\Phi}_J \mathbf{x}\|_{\mathbf{D}^{1/2}}^2 \right)$$

$$\leq d_{\max} \|\mathbf{x}\|_{\mathbf{D}^{1/2}}^2$$

$$\leq \frac{d_{\max}}{d_{\min}} \|\mathbf{x}\|_2^2,$$

which establishes the upper frame bound (i.e., the rightmost inequality). Similarly, to establish the lower frame bound, we have

$$\sum_{j=0}^{J} \|\mathbf{\Psi}_j \mathbf{x}\|_2^2 + \|\mathbf{\Phi}_J \mathbf{x}\|_2^2 \geq d_{\min} \left( \sum_{j=0}^{J} \|\mathbf{\Psi}_j \mathbf{x}\|_{\mathbf{D}^{1/2}}^2 + \|\mathbf{\Phi}_J \mathbf{x}\|_{\mathbf{D}^{1/2}}^2 \right)$$

$$\geq c d_{\min} \|\mathbf{x}\|_{\mathbf{D}^{1/2}}^2$$

$$\geq c \frac{d_{\min}}{d_{\max}} \|\mathbf{x}\|_2^2.$$

$\square$

*The proof of Lemma A.1.* Let $\mathbf{w} : V \to \mathbb{R}^D$, written in vector form as

$$\mathbf{w} = [\mathbf{w}[1]^\top, \dots, \mathbf{w}[n]^T]^\top$$
$$= [\mathbf{w}[1][1], \dots, \mathbf{w}[1][D], \mathbf{w}[2][1], \dots, \mathbf{w}[2][D], \dots, \mathbf{w}[n][1], \dots, \mathbf{w}[n][D]]^\top.$$

We first observe that we may write $\mathbf{Q}' = \mathbf{P} \otimes \mathbf{I}$, where $\mathbf{I}$ is the $D \times D$ identity matrix and $\otimes$ denotes the Kronecker product. By the mixed-product property of the Kronecker product, this implies that for all $m \geq 0$, we have

$$(\mathbf{Q}')^m = \mathbf{P}^m \otimes \mathbf{I}.$$

Therefore, by linearity, we have

$$\widetilde{\mathbf{\Psi}}_j' = \mathbf{\Psi}_j \otimes \mathbf{I},$$

and similarly $\widetilde{\mathbf{\Phi}}_J' = \mathbf{\Phi}_J \otimes \mathbf{I}$.

Now, let $\mathbf{\Pi} \in \mathbb{R}^{nD \times nD}$ be the permutation matrix such that applying $\mathbf{\Pi}$ to $\mathbf{w}$ yields

$$\mathbf{\Pi w} = [\mathbf{w}[1][1], \dots, \mathbf{w}[n][1], \mathbf{w}[1][2], \dots, \mathbf{w}[n][2], \dots, \mathbf{w}[1][D], \dots, \mathbf{w}[n][D]]^\top$$
$$= [\mathbf{w}[:][1]^\top, \mathbf{w}[:][1]^\top, \dots, \mathbf{w}[:][D]^\top]^\top,$$

where $\mathbf{w}[:][k]^\top = [\mathbf{w}[1][k], \ldots, \mathbf{w}[n][k]]^\top$ for all $1 \leq k \leq d$. Formally, let $\mathbf{\Pi}$ be the matrix corresponding to the permutation $\widetilde{\sigma} : \{1, 2, 3, \ldots, nd\} \to \{1, 2, 3, \ldots, nd\}, \widetilde{\sigma}(i) = n \cdot ((i-1) \mod d) + \lceil \frac{j}{d} \rceil$. To understand this permutation, note that we have reorder the entries of the vector $\mathbf{w}$ so that all entries corresponding to the first output dimension of the function $\mathbf{w} : V \to \mathbb{R}^d$ come first, then all of the entries corresponding to the second output dimension come next, etc.

Since $\mathbf{Q}' = \mathbf{P} \otimes \mathbf{I}$, applying $\mathbf{\Pi}$ to both the columns and the rows of $\mathbf{Q}'$ yields

$$\mathbf{\Pi}\mathbf{Q}'\mathbf{\Pi}^\top = \mathbf{I} \otimes \mathbf{P} = \begin{pmatrix} \mathbf{P} & 0 & 0 & 0 & 0 & 0 \\ 0 & \mathbf{P} & 0 & 0 & 0 & 0 \\ \ddots & \ddots & \ddots & \ddots & \ddots & \ddots \\ \ddots & \ddots & \ddots & \ddots & \ddots & \ddots \\ 0 & 0 & 0 & 0 & 0 & \mathbf{P} \end{pmatrix}.$$

Moreover, since $\mathbf{\Pi}^\top \mathbf{\Pi} = \mathbf{I}$, we see that

$$(\mathbf{\Pi}\mathbf{Q}'\mathbf{\Pi}^\top)^m = \mathbf{\Pi}(\mathbf{Q}')^m\mathbf{\Pi}^\top = \begin{pmatrix} \mathbf{P}^m & 0 & 0 & 0 & 0 & 0 \\ 0 & \mathbf{P}^m & 0 & 0 & 0 & 0 \\ \ddots & \ddots & \ddots & \ddots & \ddots & \ddots \\ \ddots & \ddots & \ddots & \ddots & \ddots & \ddots \\ 0 & 0 & 0 & 0 & 0 & \mathbf{P}^m \end{pmatrix} = \mathbf{I} \otimes \mathbf{P}^m.$$

for all $m \geq 0$, and so

$$\mathbf{\Pi}\widetilde{\mathbf{\Psi}}'_j\mathbf{\Pi} = \mathbf{I} \otimes \mathbf{\Psi}_j = \begin{pmatrix} \mathbf{\Psi}_j & 0 & 0 & 0 & 0 & 0 \\ 0 & \mathbf{\Psi}_j & 0 & 0 & 0 & 0 \\ \ddots & \ddots & \ddots & \ddots & \ddots & \ddots \\ \ddots & \ddots & \ddots & \ddots & \ddots & \ddots \\ 0 & 0 & 0 & 0 & 0 & \mathbf{\Psi}_j \end{pmatrix},$$

and similarly, $\mathbf{\Pi}\widetilde{\mathbf{\Phi}}'_J\mathbf{\Pi} = \mathbf{I} \otimes \mathbf{\Phi}_J$. Therefore, we have

$$\mathbf{\Pi}\widetilde{\mathbf{\Psi}}'_j\mathbf{w} = (\mathbf{\Pi}\widetilde{\mathbf{\Psi}}'_j\mathbf{\Pi}^\top)(\mathbf{\Pi}\mathbf{w}) = [\mathbf{\Psi}_j\mathbf{w}[:][1], \mathbf{\Psi}_j\mathbf{w}[:][2], \ldots, \mathbf{\Psi}_j\mathbf{w}][: [D]]^\top,$$

with an analogous equation for $\widetilde{\mathbf{\Phi}}'_J$. Thus,

$$\begin{aligned}
\sum_{j=0}^{J} \|\widetilde{\mathbf{\Psi}}'_j\mathbf{w}\|^2 + \|\widetilde{\mathbf{\Phi}}'_J\mathbf{w}\|^2 &= \sum_{j=0}^{J} \|\mathbf{\Pi}\widetilde{\mathbf{\Psi}}'_j\mathbf{w}\|_2^2 + \|\widetilde{\mathbf{\Phi}}'_J\mathbf{w}\|_2^2 \\
&= \sum_{j=0}^{J}\sum_{k=1}^{D} \|\mathbf{\Psi}_j\mathbf{w}[:][k]\|_2^2 + \sum_{k=1}^{D} \|\mathbf{\Phi}_J\mathbf{w}[:][k]\|_2^2 \\
&= \sum_{k=1}^{D} \left( \sum_{j=0}^{J} \|\mathbf{\Psi}_j\mathbf{w}[:][k]\|_2^2 + \|\mathbf{\Phi}_J\mathbf{w}[:][k]\|_2^2 \right).
\end{aligned} \tag{10}$$

By Corollary B.2, we have

$$c\frac{d_{\min}}{d_{\max}}\|\mathbf{w}[:][k]\|_2^2 \leq \sum_{j=0}^{J} \|\mathbf{\Psi}_j\mathbf{w}[:][k]\|_2^2 + \|\mathbf{\Phi}_J\mathbf{w}[:][k]\|_2^2 \leq \frac{d_{\max}}{d_{\min}}\|\mathbf{w}[:][k]\|_2^2,$$

and so, using (10) together with the fact that $\|\mathbf{w}\|_2^2 = \sum_{k=1}^{D} \|\mathbf{w}[:][k]\|_2^2$, we have

$$
\begin{aligned}
c\frac{d_{\min}}{d_{\max}}\|\mathbf{w}\|_2^2 &= c\frac{d_{\min}}{d_{\max}} \sum_{k=1}^{D} \|\mathbf{w}[:][k]\|_2^2 \\
&\leq \sum_{k=1}^{D} \left( \sum_{j=0}^{J} \|\mathbf{\Psi}_j \mathbf{w}[:][k]\|_2^2 + \|\mathbf{\Phi}_J \mathbf{w}[:][k]\|_2^2 \right) \\
&= \sum_{j=0}^{J} \|\widetilde{\mathbf{\Psi}}_j' \mathbf{w}\|^2 + \|\widetilde{\mathbf{\Phi}}_J' \mathbf{w}\|^2 \\
&= \sum_{k=1}^{D} \left( \sum_{j=0}^{J} \|\mathbf{\Psi}_j \mathbf{w}[:][k]\|_2^2 + \|\mathbf{\Phi}_J \mathbf{w}[:][k]\|_2^2 \right) \\
&\leq \sum_{k=1}^{D} \frac{d_{\max}}{d_{\min}} \|\mathbf{w}[:][k]\|_2^2 \\
&= \frac{d_{\max}}{d_{\min}} \|\mathbf{w}\|_2^2. \qquad \qquad \square
\end{aligned}
$$

## B.2. The Proof of Lemma A.2

*Proof.* When $m = 0$, this follows from the fact that for all $i$, $\mathcal{O}_{i,i} = \mathbf{U}_i \mathbf{U}_i^\top = \mathbf{I}$ since $\mathbf{U}_i$ is unitary, and so $\mathbf{Q}^0 = \mathbf{I} = \mathbf{I}\mathbf{I} = \mathbf{P}^0 \mathcal{O}_{i,i}$. The case $m = 1$ follows directly from the definition of $\mathbf{Q}$.

Now, reasoning by induction, assume the result holds for $m$. Then, using block matrix multiplication, we have

$$
\begin{aligned}
\mathbf{Q}^{m+1}[i,j] &= \sum_{k=1}^{n} \mathbf{Q}^m[i,k]\mathbf{Q}[k,j] \\
&= \sum_{k=1}^{n} \mathbf{P}^m[i,k]\mathbf{P}[k,j]\mathcal{O}_{i,k}\mathcal{O}_{k,j} \\
&= \sum_{k=1}^{n} \mathbf{P}^m[i,k]\mathbf{P}[k,j]\mathbf{U}_i \mathbf{U}_k^\top \mathbf{U}_k \mathbf{U}_j^\top \\
&= \sum_{k=1}^{n} \mathbf{P}^m[i,k]\mathbf{P}[k,j]\mathbf{U}_i \mathbf{U}_j^\top \\
&= \sum_{k=1}^{n} \mathbf{P}^m[i,k]\mathbf{P}[k,j]\mathcal{O}_{i,j} \\
&= \mathbf{P}^{m+1}[i,j]\mathcal{O}_{i,j}.
\end{aligned}
$$

$\square$

## B.3. The proof of Lemma A.3

We first prove the following auxiliary lemma.

**Proposition B.3.** $\overline{\mathcal{O}}_{i,j}$, *the change of coordinate matrix in the rotated coordinate system, satisfies*

$$
\overline{\mathcal{O}}_{i,j} = \mathbf{R}\mathcal{O}_{i,j}\mathbf{R}^\top.
$$

*Therefore, the corresponding vector-valued diffusion matrix* $\overline{\mathbf{Q}}$ *can be written in block form by*

$$
\overline{\mathbf{Q}}[i,j] = \mathbf{P}[i,j]\mathbf{R}\mathcal{O}_{i,j}\mathbf{R}^\top.
$$

*Proof.* Since $\overline{v}_i = \mathbf{R}v_i$, we have that $\overline{v}_i - \overline{v}_j = \mathbf{R}(v_i - v_j)$. This implies $\overline{\mathbf{C}}_i = \mathbf{R}\mathbf{C}_i$ and $\overline{\mathbf{D}_i} = \mathbf{D}_i$ (since rotation preserves distances), which implies

$$\overline{\mathbf{B}}_i = \mathbf{R}\mathbf{B}_i.$$

Therefore, $\overline{\mathbf{B}}_i = \mathbf{R}\mathbf{U}_i\mathbf{\Sigma}_i\mathbf{V}_i^\top$ is a singular value decomposition of $\overline{\mathbf{B}}_i$ and so we have $\overline{\mathbf{U}}_i = \mathbf{R}\mathbf{U}_i$. This implies that

$$\overline{\mathcal{O}}_{i,j} = \overline{\mathbf{U}}_i\overline{\mathbf{U}}_j^\top = (\mathbf{R}\mathbf{U}_i)(\mathbf{R}\mathbf{U}_j)^\top = \mathbf{R}\mathbf{U}_i\mathbf{U}_j^\top\mathbf{R}^\top = \mathbf{R}\mathcal{O}_{i,j}\mathbf{R}^\top,$$

which proves the first claim. The second claim follows immediately recalling that we assume that the edge-connectivity is unchanged by the rotation and so $\overline{\mathbf{P}} = \mathbf{P}$. $\square$

We now prove Lemma A.3.

*Proof.* In the case where $m = 0$, the result simply states that $\overline{\mathbf{w}} = \mathbf{R} \cdot \mathbf{w}$, which is true by the assumption that rotating the system rotates the vector-valued node features (vertex by vertex).

For $m \geq 1$, we use induction. In the base case, $m = 1$, we use the definition of block-matrix multiplication, as well as Lemma B.3 to write

$$\begin{aligned}
(\overline{\mathbf{Q}}\overline{\mathbf{w}})[i] &= (\overline{\mathbf{Q}}(\mathbf{R} \cdot \mathbf{w}))[i] \\
&= \sum_{j=1}^n \overline{\mathbf{Q}}[i,j](\mathbf{R} \cdot \mathbf{w})[j] \\
&= \sum_{j=1}^n \mathbf{P}[i,j]\mathbf{R}\mathcal{O}_{i,j}\mathbf{R}^\top\mathbf{R}\mathbf{w}[j] \\
&= \mathbf{R}\sum_{j=1}^n \mathbf{P}[i,j]\mathcal{O}_{i,j}\mathbf{w}[j] \\
&= \mathbf{R}(\mathbf{Q}\mathbf{w}[i]).
\end{aligned}$$

Since $i$ was arbitrary, this implies $\overline{\mathbf{Q}}\overline{\mathbf{w}} = \mathbf{R} \cdot (\mathbf{Q}\mathbf{w})$ and establishes the base case.

We now assume the result holds for some $m \geq 1$. We let $\mathbf{y} = \mathbf{Q}^m\mathbf{w}$ and let $\overline{\mathbf{y}} = \overline{\mathbf{Q}}^m\overline{\mathbf{w}}$. Observe that $\overline{\mathbf{y}} = \mathbf{R} \cdot (\mathbf{Q}^m\mathbf{w}) = \mathbf{R} \cdot \mathbf{y}$ by the inductive hypothesis. Thus,

$$\begin{aligned}
\overline{\mathbf{Q}}^{m+1}\overline{\mathbf{w}} &= \overline{\mathbf{Q}}\overline{\mathbf{y}} \\
&= \overline{\mathbf{Q}}(\mathbf{R} \cdot \mathbf{y}) \\
&= \mathbf{R} \cdot (\mathbf{Q}\mathbf{y}) \\
&= \mathbf{R} \cdot (\mathbf{Q}\mathbf{Q}^m\mathbf{w}) \\
&= \mathbf{R} \cdot \mathbf{Q}^{m+1}\mathbf{w},
\end{aligned} \tag{11}$$

where in (11) we use the inductive hypothesis with $\mathbf{y}$ in place of $\mathbf{w}$.

$\square$

## C. Removing the sign ambiguity

The definition of diffusion wavelets relies on computing the SVD of the matrices $\mathbf{B}_i$. However, as noted in Section 3, the SVD is only unique up to sign-flips. That is, one may obtain a different SVD $\mathbf{B}'_i = \mathbf{U}'_i\mathbf{\Sigma}_i(\mathbf{V}')_i^\top$ by replacing both $\mathbf{u}_{i,k}$ and $\mathbf{v}_{i,k}$ with $\mathbf{u}'_{i,k} = -\mathbf{u}_{i,k}$ and $\mathbf{v}'_{i,k} = -\mathbf{v}_{i,k}$ for any fixed $k$. Therefore, we utilize the sign-flipping trick described below in order to ensure that each of the $\mathcal{O}_{i,j}$ are well defined (and do not suffer from any sign ambiguity).

We first observe that the $k, \ell$-th entry of $\mathcal{O}_{i,j}$ is given by

$$\mathcal{O}_{i,j}[k,l] = \sum_{m=1}^d \mathbf{U}_i[k,m]\mathbf{U}_j^\top[m,\ell] = \sum_{m=1}^d \mathbf{U}_i[k,m]\mathbf{U}_j[\ell,m] = \langle\mathbf{u}_{i,k}, \mathbf{u}_{j,\ell}\rangle.$$

Therefore, if we are able to ensure that $\langle \mathbf{u}_{i,k}, \mathbf{u}_{j,\ell} \rangle$ is always non-negative, then the definition of $\mathcal{O}_{i,j}$ does not suffer from the sign ambiguity.

Accordingly, when computing $\mathcal{O}_{i,j}$, we check whether or not each $\langle \mathbf{u}_{i,k}, \mathbf{u}_{j,\ell} \rangle$ is positive or negative. If $\langle \mathbf{u}_{i,k}, \mathbf{u}_{j,\ell} \rangle$ is negative, we replace $\mathbf{u}_{j,\ell}$ with $-\mathbf{u}_{j,\ell}$. We note that in this construction, the individual vectors, $\mathbf{u}_{i,k}$ still do suffer from the sign-flip ambiguity. However, if one replaces $\mathbf{u}_{i,k}$ with $-\mathbf{u}_{i,k}$, the sign flipping trick also forces us to replace $\mathbf{u}_{j,\ell}$ with $-\mathbf{u}_{j,\ell}$. Thus, the inner products stay the same since $\langle \mathbf{u}_{i,k}, \mathbf{u}_{j,\ell} \rangle = \langle -\mathbf{u}_{i,k}, -\mathbf{u}_{j,\ell} \rangle$.

## D. Complexity of diffusion wavelets

We compute the scalar diffusion matrix $\mathbf{P}$ and the vector diffusion matrix $\mathbf{Q}$ for each graph as a one-time data preprocessing step and cache their (sparse) tensor data in memory. Since operators are independent for each graph, for large data sets this is a parallel task, so we also batch-parallelize this step across CPU workers.

For a single graph with $n$ nodes, $m$ edges, vector dimension $D$, and individual node degrees $k_i$:

- Constructing $\mathbf{P}$ as a sparse matrix involves building a sparse adjacency matrix, computing a node degree vector, adding a sparse identity matrix and rescaling entries. The total time complexity is $\mathcal{O}(n+m)$, and $\mathbf{P}$ has $n+m$ nonzero entries.

- Constructing $\mathbf{Q}$ as a sparse block-diagonal matrix requires first building a dictionary of neighbor sets ($\mathcal{O}(m)$), and then centering and kernel-rescaling vector node features ($\mathcal{O}(mD)$). Then, singular value decomposition is performed on each transformed node's vector feature matrix $C_i \in \mathbb{R}^{d \times k_i}$, which has complexity $\mathcal{O}(D \sum_{i=1}^{n} k_i^2)$ where $k_i \geq D$. Finally, computing $O_{ij} = O_i O_j^\top$ (with sign-alignment of nodes' left singular vectors) and rescaling by $p_{ij}$ is $\mathcal{O}((n+m)D^3)$. The number of nonzero entries in $\mathbf{Q}$ is $(n+m)D^2$.

These constructions of $\mathbf{P}$ and $\mathbf{Q}$ enable highly efficient geometric scattering layers in VDW-GNN, through sparse matrix multiplication of (dense) scalar and vector feature vectors. Still, in terms of complexity of the forward pass of our model, this layer dominates for typical graphs. For either a scalar or vector track, first-order scattering has complexity $\mathcal{O}(S\,m\,(F+D))$, where $S$ is the number of wavelets used (typically four to ten), $F$ is the number of signal channels (generally, $F = 1$ for a vector track), and $D$ is the dimension ($D = 1$ for scalars). To obtain second-order scattering coefficients, we recursively reapply each (sparse) diffusion operator to all first-order scattering coefficients, and then discard coefficients resulting from higher-pass wavelets reapplied to lower-pass wavelets. Accordingly, for either track, second-order scattering has complexity $\mathcal{O}(n\,F\,D\,S^2)$, where $n$ is the number of nodes.

Our codebase and vector diffusion modules are implemented in PyTorch (Ansel et al., 2024), PyTorch-Geometric (Fey & Lenssen, 2019). As such, our method implementations exploit the sparse tensor libraries in these frameworks. Finally, for all experiments, each model training run was done on one NVIDIA L40S 48 GB GPU with CUDA 12.4.

## E. Stability of local PCA and convergence of VDWs

As noted in Footnote 3, VDWs rely on local PCA to estimate a basis for the tangent space at each point. This estimate naturally improves with sufficient sampling density around a point. But, if the number of neighbors for a node is less than the vector feature dimension ($n_i < D$), then $\mathbf{U}$ will be rank-deficient, and we can't recover a full basis for that node's tangent space via local PCA. In such cases, our method requires adding neighbors until $n_i \geq D$. However, in low-density regions, where neighbors are few and far between, this may result in a poor estimate.

In such cases, it may be useful to modify the construction of $\mathbf{Q}$ to down-weight the influence of their nodes. (This did not end up being necessary for the datasets used in our experiments.) Alternatively, one could include more points beyond immediate neighbors. For instance, one could use all points $v_j$ where $\|v_j - v_i\|_2 \leq \delta_i$ (with $\delta_i$ being a threshold that accounts for sampling density), analogous to the CkNN graphs considered in Berry & Sauer (2019). Additionally, one could attempt to improve stability via the use robust tangent space estimators such as those introduced in Kohli et al. or Zhan & Yin (2011).

Our method, in particular its use of the vector diffusion operator $\mathbf{Q}$, is motivated in part by the manifold hypothesis, i.e., the assumption that the data points lie on or near a low-dimensional manifold. Under this assumption, several works such as Chew et al. (2024); Johnson et al. (2025a); Wang et al. (2023) have analyzed the convergence of graph neural networks

(with scalar-valued signals) to a continuum limit as $n \to \infty$. It is natural to ask if these results extend to our setting of vector-valued signals. As noted in Section 2.2, we view powers of the vector diffusion operator $\mathbf{Q}$ as a computationally efficient proxy for the heat semigroup $e^{-t\Delta}$ associated with the connection Laplacian $\Delta$. The convergence of networks constructed from this heat semigroup was analyzed in Theorem 1 of Battiloro et al. (2024). This analysis shows that VDW-GNNs are guaranteed to converge (in probability, with learnable parameters held constant) if we use a modified family of wavelets defined via the heat-semigroup, e.g., replace $\mathbf{Q}^{2^{j-1}} - \mathbf{Q}^{2^j}$ with $e^{-2^{j-1}\Delta} - e^{-2^j\Delta}$ (defined in the spectral domain). However, in our implementation we prefer to use wavelets defined as powers of $\mathbf{Q}$ since it allows the wavelet transform to be computed via sparse matrix-vector multiplications, and thereby increases the computational efficiency of our method. It may be possible to analyze the convergence of VDW-GNNs in this setting using the techniques introduce in Singer & Wu (2017). However, we leave that as an avenue for future work.

Lastly, we note that in general, we use $k$-NN graphs when constructing VDWs. For analysis of how the choice of $k$ can affect convergence of neural networks under the manifold hypothesis, we refer the reader to Wang et al. (2025) or Johnson et al. (2025a) and the references therein. However, in our implementation, our specific choices of $k$ were guided by (i) the aim of inducing sparsity in order to increase computational efficiency and scalability (we prioritize small $k$ where possible, challenging all models to work with sparse graphs), and (ii) tuning for the task: for example, in the wind vector reconstruction task (Section 4.2), interpolated vectors appeared to be most similar to their immediate neighbors' known vectors, making the highly local signal most important, captured best with $k = 3$.

# F. Experimental details - Ellipsoids (Section 4.1)

## F.1. Architecture

In this section, we provide details on the VDW-GNN model that we used for our ellipsoids experiments in Section 4.1. A graphical illustration of this model is given in Figure 2 of the main text.

**Scalar- and vector-track geometric scattering layers.** We first separately compute the first- and second-order scattering coefficients of each input signal, using (3) and (4) for the scalar-valued signals and (6) and (7) for the vector-valued signal, and concatenate these to the zeroth-order scattering coefficients (where the zeroth-order coefficients are simply the untransformed signals). We organize the scalar-valued coefficients and vector-valued scattering coefficients into tensors $\widetilde{\mathbf{X}}$ and $\widetilde{\mathbf{W}}$ with dimensions $n \times F_{\text{scalar}} \times S$ and $n \times D \times S$, where $S$ is the total number of zeroth-, first-, and second-order used for each signal at each node. (Note that for simplicity, we generally assume that we are given a single vector-valued signal, i.e., $F_{\text{vector}} = 1$, which is the case for all of our experiments.) We note that in our experiments, we use the same value of $S$ for all vector- and scalar-valued features. However, we could readily modify our architecture to use a different number of scales for each feature.

**Within-track mixing layers.** Second, for each track, the scattering coefficients are passed through within-track coefficient-mixing layers. For the scalar track, we learn new combinations of the scattering scales using a two-layer MLP, resulting in a new tensor $\widetilde{\mathbf{X}}' \in \mathbb{R}^{n \times F_{\text{scalar}} \times K}$ defined by

$$\widetilde{\mathbf{X}}'[i_1, i_2, :] = \text{MLP}(\widetilde{\mathbf{X}}[i_1, i_2, :]), \tag{12}$$

(where the MLP does not depend on $i_1$ or $i_2$).

For the vector track, we perform a weighted summation across the signals with learnable weights and also employ a gating procedure. This yields a new tensor $\widetilde{\mathbf{W}}' \in \mathbb{R}^{n \times d \times K}$ defined by

$$\widetilde{\mathbf{W}}'[i_1, i_2, i_3] = \sigma_w(\alpha_{i_3}) \sum_{s=1}^{S} \theta_{i_3, \ell} \widetilde{\mathbf{W}}[i_1, i_2, s], \tag{13}$$

where each $\alpha_{i_3} \in \mathbb{R}$ is a learnable gating parameter and $\sigma_w$ is a nonlinear activation.

**Vector invariants extraction layer.** Third, for the vector track, we extract three invariant scattering vector features, the norm of each vector as well as mean and maximal cosine similarity between each of these vectors and those of its neighboring nodes. We then concatenate these invariant vector features with the scalar track hidden representations into a combined invariant hidden feature tensor $\mathbf{T} \in \mathbb{R}^{n \times (F_{\text{scalar}} + 3) \times K}$. For each node $v_i$, we then reshape $\mathbf{T}[i, :, :]$ to obtain a flattened hidden feature vector $\mathbf{t}_i \in \mathbb{R}^{K(F_{\text{scalar}} + 3)}$.

**Task-specific prediction head.** For scalar-valued target functions, we design our network to be rotationally invariant. Therefore, for node-level tasks, we use a five-layer MLP to map each $\mathbf{t}_i$ to the target dimension. Alternatively, for graph-level tasks with scalar targets, we first pool the vectors $\mathbf{t}_i$ vectors across nodes within graphs (using, e.g., summation over $i$ or max aggregations over $i$), and again use a five-layer MLP as the final readout head.

For vector-valued target functions, we seek for our network to be rotationally equivariant. Thus, our final prediction for each node $v_i$ is given by

$$\widehat{\mathbf{y}}_i = \sum_{k=1}^{K} \sigma_w(\beta_i[k]) \, \widetilde{\mathbf{W}}'[i, :, k] \in \mathbb{R}^D \,,$$

where, as above $\sigma_w$ is a nonlinear activation and each $\beta_i[k]$ is a gating parameter. Here, however, the gating parameters $\beta_i[k]$ are learned via a two-layer MLP that inputs the tensor $\mathbf{T}[i, :, :]$ and outputs $\boldsymbol{\beta}_i$, a vector of $K$ gating weights for each node.

**Further details.** We also note that our implementation of the scattering coefficients differs slightly from the exposition in Section 3 for improved performance. Instead of dyadic wavelets of the form $\mathbf{P}^{2^{j-1}} - \mathbf{P}^{2^j}$, we instead use generalized diffusion wavelets of the form $\mathbf{P}^{t_{j-1}} - \mathbf{P}^{t_j}$, where $0 = t_0 < t_1 < \ldots, < t_J$ is an increasing sequence of diffusion scales which are selected by the InfoGain procedure introduced in Johnson et al. (2025b). Additionally, we did not use a non-linearity while computing the scattering-coefficients since our network is able to learn non-linear relationships in the data via the MLP and the gating function used in (12) and (13). Following Bhaskar et al. (2025), in order to help the geometric scattering transform probe the graph geometry, we also used two Dirac (Kronecker) vectors as additional scalar-valued input signals. To place these Diracs, we first computed the centroid of the vertices in $\mathbb{R}^d$ (using the Euclidean distance). We then placed one Dirac at the vertex closest to this centroid and the other Dirac at the vertex furthest from the centroid (ties broken randomly). Importantly, we note that all of our theoretical guarantees may be readily adapted to this modified version of the vector-valued geometric scattering transform.

### F.2. Cross-validation procedure

In our experiments, we split the data into five folds (each containing 20% of the data), and perform five-fold cross validation such that each fold is used as a test set and validation set exactly once, while the remaining three folds make up the training set in each cross-validation step.

### F.3. Hyperparameter settings

#### F.3.1. GENERAL

For all models, we optimize for mean squared error (MSE) loss using PyTorch's `AdamW` optimizer, an initial learning rate of 0.001, and a train batch size of 32 (except for TFN, for which we use 16, to prevent out-of-memory errors). Models train for at least 50 burn-in epochs, before the following early stopping is enforced: if validation loss does not improve for 50 consecutive epochs (checking every five epochs), we (1) halve the learning rate and reload the best model weights achieved (by validation loss), and then (2) quit training after a maximum of two such restarts or a maximum of 500 total training epochs. The final model weights are then extracted from the epoch in which the lowest validation loss was achieved.

For experimental consistency and compatibility with our regression tasks, we standardize some modules across models as appropriate. That is, first, for the graph-level diameter estimation task, we use `sum` and `max` pooling of hidden node features wherever such pooling is needed; for VDW-GNN, LEGS, GCN, GIN, and GAT, these pooled features are input into a five-layer readout MLP with hidden dimensions 128, 64, 32, and 16. Second, for all models (including ours), we use the sigmoid linear unit (SiLU, or 'swish') activation function (Elfwing et al., 2018) in the MLP (except for when another activation function is explicitly prescribed by the baseline model.) Third, when edge weights are used by a model, we compute these weights using a Gaussian kernel $K_\epsilon$ from Section 3, with scale parameter $\epsilon$ set equal to the squared mean of mean neighbor distances across the final ellipsoid data set for all models.

#### F.3.2. VDW-GNN (AND VDW-GNN-NON-EQUIVARIANT)

For VDW-GNN, when using the InfoGain Wavelets procedure to select wavelet diffusion scales, we set $t_J = 16$, and use information cutoff quantiles of $[0.25, 0.5, 0.75]$ (see Johnson et al. (2025b) for details). We deviated slightly from the method presented in Johnson et al. (2025b) and measured the convergence of each feature as quantified in the $\ell^1$ norm rather than the KL divergence. For the scalar signals, we applied the $n \times n$ diffusion matrix $\mathbf{P}$ to each scalar signal $\mathbf{x}$.

For the vector signals, we applied the $nD \times nD$ vector diffusion matrix $\mathbf{Q}$ to each vector-valued signal $\mathbf{w} \in \mathbb{R}^{nd}$. This procedure produced diffusion scales $\{t_j\}_{j=0}^J = \{0, 1, 2, 4, 6, 8, 16\}$ for the scalar features ) and $\{0, 1, 2, 4, 6, 9, 16\}$ for the vector features (after taking the median across input signals as applicable). This means that that our filter bank consists seven total filters (including the low-pass filter $\mathbf{\Psi}^{t_J}$). In the within-track mixing layers, we set $K$ equal to 16 and 32 for the scalar and vector tracks, respectively, and the hidden layer dimensions of their two-layer mixing MLPs to $[64, 64]$ and $[128, 128]$. (Note the vector track mixing MLP has no biases or activations.) We use sigmoid for the gate activation $\sigma_w$.

On the graph-level diameter estimation task, we mitigate overfitting by introducing random perceptron dropout between layers of the readout MLP, with probability 0.7. On the node-level vector target regression task, the final vector gating coefficients MLP has layer widths $[128, 128, K]$, including the output layer, and SiLU activations. As before, we apply sigmoid for the gates' activation function, $\sigma_w$.

When ablating the vector track in VDW-GNN to render it non-equivariant, we (1) concatenate the vector feature coordinates to the Dirac scalar features as $D$ separate, additional scalar features, and (2) omit all vector feature operations. (For vector targets, this ablated model then uses the five-layer MLP prediction head defined previously, with an output layer width of the vector dimension $D$.)

### F.3.3. LEGS

For LEGS, we use Dirac scalar node features, as in VDW-GNN, and concatenate the vector feature coordinates as $d$ additional scalar features (analogous to VDW-GNN-non-equivariant). The network employs dyadic-scale wavelets with $J = 4$. Hidden representations are fed into a five-layer MLP with SiLU activations (and no dropout) to produce predictions. For graph-level predictions, we first pool across nodes using both `sum` and `max` operators. For vector targets, the prediction head outputs a vector of dimension $d$.

### F.3.4. GCN, GIN, GAT

For GCN, GIN, and GAT, the input features consist of the vector feature coordinates encoded as $D$ scalar features. Each model uses two layers, with hidden dimensions size 128. Predictions are obtained using the same MLP head strategy as in LEGS.

### F.3.5. EGNN AND TFN

For the EGNN and TFN models, we adapted code from the "Geometric GNN Dojo" (Joshi et al., 2023) under the MIT license, from their Github repository (`https://github.com/chaitjo/geometric-gnn-dojo`). For these models, on the node-level equivariant regression task, we tune the number of layers separately for each task by starting from one and increasing until performance degrades rather than improves. The best-performing configurations use two layers for EGNN and four layers for TFN on the graph-level task; and seven and four layers, respectively, on the node-level task. Prediction heads are implemented as two-layer MLPs (which operate on both `sum` and `max` pooled hidden node features for graph-level predictions).

EGNN expects a scalar node feature embedding; here, we use a single uniform feature with embedding size 128, which is equivalent to a learnable bias. TFN requires specification of the maximum irreducible representation order $\ell$, which we tune up to $\ell = 2$ to prevent excessive parameter growth. We find that $\ell = 2$ was best for both tasks. TFN also expects radial edge weights; to this end, we implement Gaussian kernel edge weights, one per edge, for use in message passing.

### F.4. Details on the node-level vector target

The direction of $\mathbf{h}(v)$ at each point $v = (x, y, z)$ is chosen to be the outward normal vectors, which is given by the normalized gradient of the functions $f(x, y, z) = x^2/a^2 + y^2/b^2 + z^2/c^2$, i.e.,

$$\mathbf{n} = \frac{\nabla f(v)}{\|\nabla f(v)\|}, \qquad \nabla f(v) = 2\left(\tfrac{x}{a^2}, \tfrac{y}{b^2}, \tfrac{z}{c^2}\right).$$

The magnitude of $\mathbf{h}(v)$ is computed using (an approximation of) the $K = 16$ nontrivial eigenfunctions of the Laplace-Beltrami operator on the underlying ellipsoid. (The first eigenfuction is considered trivial because it is constant and has eigenvalue 0.) To compute these eigenfunctions, we sample 896 more points per point cloud (so that there is a total of 1024 points on each point cloud). We then construct a symmetric, unweighted, $k$-NN graph, $G^{\text{large}} = (V^{\text{large}}, E^{\text{large}})$ from

each of these enlarged point clouds (with $k = 10$) and compute the symmetric-normalized graph Laplacian $\mathbf{L}_{\text{sym}}^{\text{large}} = \mathbf{I} - (\mathbf{D}^{\text{large}})^{-1/2} \mathbf{A}^{\text{large}} (\mathbf{D}^{\text{large}})^{-1/2}$. We then approximate the first 16 non-trivial eigenfuctions of the Laplace Beltrami operator by the first eigenvectors of $\mathbf{L}_{\text{sym}}^{\text{large}}$.

We then let $\mathbf{g}$ be a real-valued signal defined by $V^{\text{large}}$ by

$$\mathbf{g}(v) = \sum_{j=1}^{K} c_j \, \phi_j(v),$$

and rescale $\mathbf{g}$ so that its entries have magnitudes in $[0, a]$, for some fixed $0 < a < 1$, by setting $\widetilde{\mathbf{g}} = a \frac{\mathbf{g}}{\|\mathbf{g}\|_\infty}$. Finally, we define the magnitude of $\mathbf{h}$ by $\|\mathbf{h}(v)\|_2 = 1 + \widetilde{\mathbf{g}}(v)$. We note that by construction, we have $0 < 1 - a \le \|\mathbf{h}(v)\|_2 \le 1 + a$ for all $v$.

When training and evaluating our network, we use only the 128 points from the original data set. The additional points were merely introduced so that the eigenvectors of the graph Laplacian would be a good approximation of the Laplace-Beltrami operator on the underlying ellipsoid.

### F.5. Model runtimes

The run times for each model are for the diameter prediction task and the node-level vector prediction task are displayed in Tables 5 and 6 respectively. We note that all models were trained using one NVIDIA L40S 48 GB GPU, with one exception. Due to its high parameter count and larger memory requirements, we trained TFN using distributed data parallel (DDP) with four such GPUs. Hence we conjecture that, if the model fit on one GPU, its runtimes would roughly quadruple.

*Table 5.* Epoch number of best validation loss and average training epoch runtime on the diameter prediction task. TFN is marked with [4 GPUs] since it was trained on multiple GPUs (while all other models used a single GPU), as explained at the beginning of section F.5.

| Model | Best epoch | Sec. per epoch |
|---|---|---|
| VDW-GNN (Ours) | $461 \pm 56$ | $0.0952 \pm 0.0008$ |
| VDW-GNN (non-equivariant) (Ours) | $96 \pm 93$ | $0.0792 \pm 0.0039$ |
| LEGS | $356 \pm 155$ | $0.0946 \pm 0.0014$ |
| GCN | $335 \pm 132$ | $0.0568 \pm 0.0009$ |
| GAT | $62 \pm 31$ | $0.0662 \pm 0.0013$ |
| GIN | $130 \pm 111$ | $0.0576 \pm 0.0004$ |
| EGNN (2-layer) | $338 \pm 178$ | $0.0856 \pm 0.0009$ |
| TFN (4-layer) [4 GPUs] | $453 \pm 72$ | $2.6965 \pm 0.0044$ |

*Table 6.* Epoch number of best validation loss and average training epoch runtime on the node-level vector prediction task. TFN is marked with [4 GPUs] since it was trained on multiple GPUs (while all other models used a single GPU), as explained at the beginning of section F.5.

| Model | Best epoch | Sec. per epoch |
|---|---|---|
| VDW-GNN (Ours) | $488 \pm 4$ | $0.0945 \pm 0.0003$ |
| VDW-GNN (non-equivariant) (Ours) | $355 \pm 184$ | $0.0793 \pm 0.0034$ |
| LEGS | $402 \pm 128$ | $0.0980 \pm 0.0016$ |
| GCN | $356 \pm 176$ | $0.0583 \pm 0.0007$ |
| GAT | $356 \pm 179$ | $0.0624 \pm 0.0004$ |
| GIN | $330 \pm 171$ | $0.0562 \pm 0.0004$ |
| EGNN (7-layer) | $325 \pm 77$ | $0.1552 \pm 0.0015$ |
| TFN (4-layer) [4 GPUs] | $364 \pm 123$ | $2.6912 \pm 0.0042$ |

## G. Experimental details - Wind field reconstruction (Section 4.2)

**Lifting 2D wind vectors to 3D.** Given horizontal surface wind components $\mathbf{w}_{\text{2D}} = (u, v)$ (eastward, northward), for a point on the unit sphere with latitude $\phi$ and longitude $\lambda$, we compute the 3D wind vector by $\mathbf{w}_{\text{3D}} = u \, \mathbf{e}_{\text{east}} + v \, \mathbf{e}_{\text{north}}$ where

$\mathbf{e}_{\text{east}} = [-\sin\lambda, \cos\lambda, 0]^\top$, and $\mathbf{e}_{\text{north}} = [-\sin\phi\cos\lambda, -\sin\phi\sin\lambda, \cos\phi]^\top$ are basis vectors for the local tangent space.

**Architecture**. Here, our VDW-GNN is a lightweight vector-only graph module that maps per-node vectors to updated vectors in each block by (1) applying vector diffusion wavelets; (2) concatenating the resulting coefficients with the ordered top-$k$ incoming neighbor-vector and edge-weight features, and (3) feeding into a shared MLP to predict a new vector for each node. In the wind configuration, a single block is used with a residual update on the original vector feature. We use a custom set of vector diffusion wavelets $\widetilde{\mathcal{W}}_J = \{(\mathbf{I} - \mathbf{Q}), (\mathbf{Q} - \mathbf{Q}^2), (\mathbf{Q}^2 - \mathbf{Q}^3), \mathbf{Q}^3\}$ (where the last is the low-pass wavelet).

**Training procedure**. We enforced the following early stopping procedure for all models: after a burn-in of at least 100 epochs, if validation MSE (calculated every epoch) does not improve for 100 epochs, halve the learning rate and continue training from the previous best model weights by validation MSE, and then cease training if validation MSE fails to improve for another 100 consecutive epoch period.

**Hyperparameter settings**. Tuning the number of layers for GIN, GAT, GCN, and EGNN, we found that a single layer was optimal for these models; and two for TFN. We found a node embedding dimension of 128 worked best for EGNN, and 16 for TFN. We used a maximum irreducible representation order of $\ell = 2$ for TFN. All models use a final three-layer prediction MLP head with hidden widths (128, 128) and output dimension 3, SiLU activations, and no batch normalization or dropout. For DD-TNN, we maintained hyperparameter settings as set by Battiloro et al. (2024) in their codebase. We trained all models with PyTorch's AdamW optimizer with $(\beta_1, \beta_2) = (0.9, 0.999)$, weight decay of $10^{-6}$, and an initial learning rate of 0.005.

## H. Experimental details - Multi-channel neural recordings (Section 4.3)

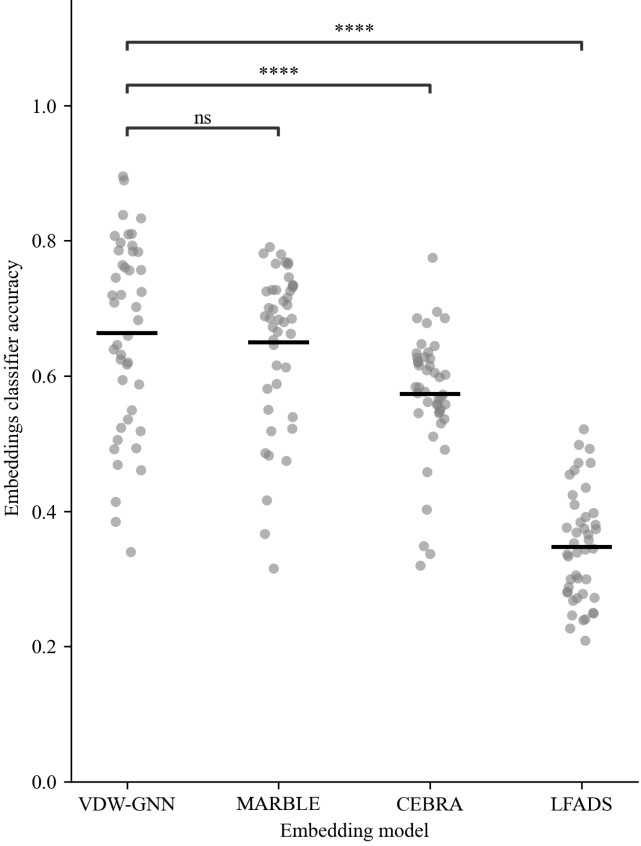

*Figure 5.* Models' distributions of mean (of mean cross-validated) classification accuracies across 44 experimental days. Brackets annotation the $p$-value results of two-sided, paired Wilcoxon tests (ns: not significant, ****: $p < 0.0001$).

**Data processing**. We use the data set version available on MARBLE's data repository (Gosztolai, 2023), which is a processed subset of the original set of experiments in Kaufman et al. (2016), and in which the raw trial recordings have been converted into rates using a Gaussian kernel (s.d. 100 ms), then subsampled to 20-millisecond intervals.[7] Following MARBLE's data processing protocol, for all models, we truncate the time series to after the 'go' cues, leaving 35 timepoints per trial, and then smooth with a Savitzky-Golay filter (order 2, window length 9). However, unlike MARBLE, we do not then apply PCA to reduce the data dimension below 24 (which we found discards useful signal, as a severe linear reduction). For our model, we follow MARBLE's lead, and reconstruct the problem as learning on a graph with vector-valued node features (note that CEBRA and LFADS do not convert the neural data to vector-valued features). Here, we follow the convention of assigning velocity vectors derived as $\mathbf{v}_t = \mathbf{x}_{t+1} - \mathbf{x}_t$ to timepoint $t$ and dropping the last timepoint, leaving a final 34 nodes per trial.

**Baselines**. MARBLE is a graph-based deep learning method that learns embeddings of neural states conceived as local flow fields on a manifold, using gradient filters and unsupervised contrastive loss. CEBRA combines nonlinear independent components analysis (ICA) and self-supervised contrastive learning to produce embeddings that jointly model neural activity data with behavioral auxiliary variables and temporal structure. LFADS is an unsupervised method built from sequential autoencoders, which instead of contrastive loss, uses a reconstruction training objective and KL divergence regularization. Additionally, we note that while MARBLE (and our model) compute and use vector-valued neural 'velocity' features, CEBRA and LFADS do not use vector-valued features, but instead ingest scalar-valued neural recording data.

**Architecture**. Our model for this task consists of three layers: (1) an equivariant, vector-valued geometric scattering transform of timepoints'/nodes' neural velocity vector features (zeroth- and first-order transforms only, using a custom set of vector diffusion wavelets $\widetilde{\mathcal{W}}_J = \{(\mathbf{I} - \mathbf{Q}), (\mathbf{Q} - \mathbf{Q}^2), (\mathbf{Q}^2 - \mathbf{Q}^3), \mathbf{Q}^3\}$ (where the last is the low-pass wavelet); (2) a four-layer MLP that projects the concatenated multi-order scattering coefficients into the embedding dimension (with structure input $\rightarrow 256, 256, 256 \rightarrow$ embedding, ReLU activations, and slight dropout $p = 0.05$); (3) a custom supervised contrastive loss function that treats nodes from the same behavioral condition as positives and nodes from different conditions as negatives, and focuses learning on informative "hard" negatives. Concretely, for each anchor embedding, we sample a fixed number of positives per anchor and then select a top-$k$ subset of negatives with highest similarity (plus optional random negatives) to form a compact but challenging contrastive batch from which to compute loss as a temperature-scaled log-softmax over these similarities that encourages tight within-condition clusters and separation across conditions.

**Inductive evaluation**. A key contribution of our experimental design is to test models on inductive (as opposed to transductive) representation learning. This evaluation protocol is more stringent than the original MARBLE setup. Whereas MARBLE trained and evaluated in a transductive manner by randomly holding out points while still permitting the test nodes to appear in the graph during embedding computation, we withhold entire trials within each day. Thus models are trained only on training trials; test trials are never used in training nor incorporated into the training graph. This trial-wise split prevents subtle data leakage via neighborhood overlap, and directly probes generalization to unseen trial trajectories.

Across all methods, we evaluate representations via an SVM probe trained on training-trial embeddings and applied to held-out trial embeddings, ensuring that the decoding stage reflects the generalization properties of the learned representations rather than the capacity of an end-to-end classifier. In our inductive protocol, entire trials are withheld from training and never included when building the training graph for our model and for MARBLE (both of which are graph-based). This creates the challenge that test nodes do not exist in the training graph at evaluation time. Rather than reconstructing an augmented graph containing test nodes (which would revert to a semi-transductive setting), we estimate test-trial embeddings using a neighbor-based distillation procedure. That is, each test node's embedding is computed as a Gaussian kernel–weighted average of the embeddings of its nearest neighbors in the training set, where neighbors are selected according to distance in neural activity space. This yields an inductive mapping from unseen nodes to the learned representation space without retraining or graph augmentation. In contrast, for CEBRA and LFADS do not use graphs or vector features. These models learn embeddings directly from trial time series, and test-trial embeddings are obtained by a forward pass through their trained encoder modules on the held-out test trials.

**Training procedure**. We employed the same learning rate adjustment and early stopping procedure as used in the wind experiments (Appendix G). However, the stopping rule is here governed by the classification accuracy achieved by the SVM (which is fit to the training set trials before each post-epoch evaluation) failing to improve for 32 epochs, after a burn-in of at least 32 epochs. For MARBLE, we preserved the learning rate reduction factor in their codebase as 0.1 (instead of 0.5, as

---

[7]In MARBLE's version of this reduced data set, some days have seven conditions, while others have eight; however, MARBLE used seven in their analysis, and we follow suit.

we used). Note that CEBRA trains with its own `fit()` or `partial_fit()` methods, and we used `partial_fit()` to enforce early stopping by checking SVM validation accuracy every 100 iterations, as one epoch.

**Hyperparameters**. For the CkNN graph used by our model and MARBLE, we used $k = 30$ neighbors and $\delta = 1.0$ (note that this creates a sparser graph than MARBLE's original $\delta = 1.4$). The support vector machine (SVM) classifier used to evaluate embeddings used a radial basis function kernel, with scale parameter $\gamma = (p\sigma)^{-1}$ (where $p$ is the number of features and $\sigma$ is the overall variance of the training data), and a regularization constant of 1.

For our supervised contrastive loss function, we used a log-softmax temperature of 0.05, and randomly sampled 128 'anchor' nodes in each epoch, and further sampled one positive-sample node, eight top-$k$ neighboring negative sample nodes, and eight random (uniformly sampled) negative sample nodes per anchor.

LFADS has a large number of hyperparameters; our tuning focused on the 'initial condition' and 'controller' encoders widths (both ultimately set to 256). Additionally, because we processed the data into smoothed, continuous-valued signals (not counts), we set its `reconstruction` parameter to "`gaussian`" instead of "`poisson`".

We trained all models with PyTorch's `AdamW` optimizer with $(\beta_1, \beta_2) = (0.9, 0.999)$, weight decay of $10^{-5}$, and an initial learning rate of 0.005.

