# OpenReview forum: "VDW-GNNs: Vector diffusion wavelets for geometric graph neural networks"
_ICML.cc/2026/Conference — ICML 2026 regular_

### Official Review · Reviewer_LU4t · 2026-03-04

**Soundness:** 3
**Presentation:** 1
**Significance:** 2
**Originality:** 3
**Overall Recommendation:** 4
**Confidence:** 3

**Summary:**

This paper proposes vector diffusion wavelets (VDWs) for graphs with vector-valued node features. The authors theoretically demonstrate that VDWs preserve frame bounds comparable to traditional diffusion wavelets and exhibit equivariance to rotations in the ambient space. The proposed wavelets are further integrated into a class of geometric graph neural networks, which achieve strong performance on synthetic point cloud datasets as well as real-world datasets, including wind-field measurements and neural activity data.

**Compliance With Llm Reviewing Policy:**

Affirmed.

**Final Justification:**

This paper presents a novel method with a reasonable experimental design. While the discussion of the experimental results is somewhat limited, the authors’ rebuttal has helped clarify several of my concerns. Taking both the paper and the rebuttal into account, my overall recommendation is Weak Accept.

**Key Questions For Authors:**

**Q0. [Explanation of questions]**

The authors are only expected to respond to the questions listed below; no response is required for the weakness section.

**Q1. [Overly strong or unsupported claims]**

(i). The following statements lack supporting references and should be properly cited to ensure the validity of the claims: Line 033, “Crucially, most GDL methods aim to ...” and “Furthermore, they aim to represent …”.

(ii). In Line 020, right column, the authors state that the number of GNN layers “must” be limited to two or three to avoid oversmoothing. This claim is overly absolute. Although oversmoothing is a known issue, several approaches, such as residual/skip connections and jumping knowledge mechanisms, have enabled deeper GNN architectures.

**Q2. [Presentation issues]**

Please pay attention to presentation issues (i.e., typos or math/grammar errors), such as:

(i). Line 070, “When convenient , will identify the signal…” should be “When convenient , we will identify the signal…”?

(ii). Line 072, $n \times F$ should be $n \times F_\mathrm{scalar}$?

(iii). Line 087, if $P^1 = P$, it is better to express this explicitly?

(iv). Line 160, “…, and we when view..” should be “…, and when we view”?

**Q3. [Insufficient discussions related to experiment results]**

Section 4.2 mainly focuses on the experiment settings and implementation of various models, whereas the discussions related to the experiment results seem to be insufficient. For example, the authors do not provide an explanation for why VDW-GNN performs significantly better on graph-level tasks than on node-level tasks. In particular, on graph-level benchmarks, VDW-GNN outperforms the second-best model by nearly an order of magnitude, whereas such a substantial advantage is not observed in the node-level experiments.

**Limitations:**

Please discuss limitations of the proposed method.

**Strengths And Weaknesses:**

**Strength**

S1. [Soundness] The proposed method is supported by theoretical proofs. Experiments include sufficient baseline models and tasks. Experimental details are clearly given.

S2. [Presentation] Figures are clearly presented and effectively support the illustrations. Mathematical notations are mostly well defined; however, the overall clarity of some mathematical expressions could be further improved.

S3. [Significance] The proposed method could contribute to the research field of GNNs, and provide direction for future research to build on.

S4. [Originality] The proposed method is a novel combination of existing techniques. This paper also provides extended method in vector space based on diffusion wavelets.

**Weakness**

W1. [Soundness - Overly strong or unsupported claims -> Q1] Some claims in the introduction is overly strong or without supported evidence.

W2. [Presentation - Presentation issues -> Q2] Many typos and grammatical errors are observed throughout the manuscript. The overall language and mathematical expressions require further polishing. At present, the readability and clarity of the paper are limited.

W3. [Significance - Insufficient discussions related to experiment results -> Q3] The discussion of the experimental results appears to be insufficient, which may limit the broader impact of the findings and the insights provided for future research. In addition, the limitations of the proposed method are not adequately analyzed.

---

> ### Author Rebuttal · Authors · 2026-03-29
>
> Re: citations — Thank you for pointing this out. To support these claims we will add a reference to [1]. We will also slightly rephrase the first claim to read “Crucially, most GDL methods aim to represent the data points in a manner that utilizes the intrinsic structure and symmetries of the data set” and, for the second, add additional references to [2, 3, 4].
>
> Re: oversmoothing — We agree that our initial phrasing should be improved. While wavelets offer one possible solution to the oversmoothing problem, there are also other viable options such as residual connections and jumping knowledge mechanisms. First, we will rephrase this sentence to clarify that this is a limitation of simple message passing networks which do not use these methods. It will read:
>
> “However, after each layer of a simple message passing network, the representation of each node becomes progressively smoother. Therefore, in such networks, the total number of layers must be kept small, typically to two or three, to avoid oversmoothing (Nt & Maehara, 2019; Qureshi et al., 2023), where the node representations become increasingly similar thus limiting their utility for machine learning tasks.”
>
> Second, we will modify the first sentence of the next paragraph to mention other options. It will read:
>
> “Several possible solutions to the oversmoothing versus underreaching tradeoff have been proposed in the literature, including the use of residual connections (He et al., 2016) and jumping knowledge mechanisms (Xu et al., 2018). In this paper, we will focus on diffusion wavelets (Coifman & Maggioni, 2006) and geometric scattering transforms (Zou & Lerman, 2019; Gama et al., 2018; Gao et al., 2019) as well as associated GNNs (Min et al., 2020; 2021; Tong et al., 2024)).”
>
> Re: typos — We thank the reviewer for their careful reading and apologize for these typos. We will make sure to fix all such errors in the camera-ready copy.
>
> Re: results discussion — Based on our results, it appears that wavelets may be particularly useful for graph-level tasks where it is important to capture global structure, such as the diameter estimation task for ellipsoids, due to the multiscale nature of the wavelet filters. However, on the node-level tasks which we consider, this advantage may be less pronounced, and likely explains the closer performance between models on the ellipsoids node-level task. Moreover, the wind-field reconstruction task seems to be highly local, which may limit the need for an extensive bank of band-pass wavelet filters. That being said, we are still competitive with the other equivariant GNNs while using significantly fewer parameters. We will add some discussion of this to the camera copy in order to improve our exposition.
>
> Re: limitations — The main limitations of our method are:
>
> 1. Because VDWs depend on local PCA estimates of tangent space bases, their reliability degrades when neighborhoods are too small or sparse to support accurate estimation of local geometry (we discuss this point, and potential remedies, in our response to question 2 from reviewer 3CCy).
>
> 2. While VDWs are typically highly parameter efficient, they may be sensitive to key hyperparameters with certain data sets. For instance, tuning choice of diffusion scales employed to construct a filter bank of VDWs, aiming to capture key signal bands in the data, is often a nontrivial but crucial step. Moreover, if the graph structure is not given, the scales may also need to be tuned in tandem with $k$ (if using a $k$-NN graph), and the edge-weighting kernel, as discussed further in our response to question 1 from reviewer jdRM.
>
> 3. Our method is motivated by the manifold hypothesis: that data lie approximately on a smooth, low-dimensional manifold. If this assumption is violated (due to noise or lack of manifold structure, etc.), performance may degrade. This is a limitation of manifold learning methods in general.
>
> We will add discussion of limitations to the camera-ready manuscript.
>
> [1] H. S. Borde & M. Bronstein. Mathematical Foundations of Geometric Deep Learning. arXiv:2508.02723, 2025.
>
> [2] Z. Li et al. On the completeness of invariant geometric deep learning models. arXiv:2402.04836, 2024.
>
> [3] Z. Wang Z, et al. Stability of neural networks on manifolds to relative perturbations. ICASSP 2022.
>
> [4] M. Perlmutter, et al. Geometric wavelet scattering networks on compact Riemannian manifolds. PMLR 2020.
>
> [5] K. He, et al. Deep residual learning for image recognition. IEEE CVPR 2016.
>
> [6] K. Xu, et al. Representation learning on graphs with jumping knowledge networks. ICLR 2018.

---

> > ### Author Rebuttal · Reviewer_LU4t · 2026-03-31
> >
> > Thank the authors for the response. I'll keep my score.

---

### Official Review · Reviewer_3CCy · 2026-03-11

**Soundness:** 3
**Presentation:** 2
**Significance:** 3
**Originality:** 3
**Overall Recommendation:** 4
**Confidence:** 3

**Summary:**

The manuscript introduces Vector Diffusion Wavelets (VDWs) and a corresponding neural network architecture termed VDW-GNNs for processing vector-valued signals defined on geometric graphs. Building on the Vector Diffusion Maps (VDM) framework, the authors construct a vector diffusion operator that incorporates local coordinate transformations between neighboring nodes. Wavelets are then defined on this operator, resulting in a multiscale representation for vector fields on graphs.

The authors further integrate these wavelets into a graph neural network architecture, enabling the model to process vector-valued features while respecting the geometric structure of the underlying manifold. The construction is shown to yield a stable frame system and exhibits equivariance with respect to rigid motions of the ambient space, including rotations and translations. Empirically, the framework is evaluated on synthetic vector fields on ellipsoids, real-world wind field data, and multi-channel neural recordings, demonstrating its capability to model structured vector-valued signals on graphs.

**Compliance With Llm Reviewing Policy:**

Affirmed.

**Key Questions For Authors:**

1) The manuscript would benefit from careful proofreading, as there are several grammatical and typographical errors throughout the text. For example, P2-L070 (“When convenient, will identify the signal …”) and P4 L-159 (“We let $w_i[j] = w_i(v_j) \in R^D$ denote the value
of the signal $w_i$ at each vertex, and we when view $w_i$ as a vector in $R^{nD}$ we ...”).

2) The local tangent spaces are estimated via PCA on neighborhood samples. However, this procedure may become unstable when the graph is sparse or when the number of neighboring points is limited. It would be helpful if the authors could discuss the robustness of this step under such conditions.

3) It is not entirely clear whether the vector diffusion operator is symmetric (or self-adjoint). Since some of the theoretical properties appear to rely on this assumption, the authors may wish to clarify whether this condition holds and under what assumptions.

4) The manuscript should provide empirical evidence that vector diffusion wavelets mitigate oversmoothing, for example by evaluating node representation similarity or performance across deeper layers compared to standard GNNs.

**Limitations:**

The experimental validation is currently limited by the small-scale nature of the synthetic and meteorological datasets. These constraints on node count and sample size may restrict the assessment of how VDW-GNNs perform on higher-order, more complex geometric structures. While the efficacy is demonstrated within the current scope, additional testing on large-scale datasets featuring diverse manifold geometries is required to confirm the framework’s robustness and scalability across broader industrial contexts.

**Strengths And Weaknesses:**

Soundness:
The submission is technically sound, presenting a principled extension of diffusion wavelets to the setting of vector-valued features on geometric graphs. However, while the mathematical framework is consistent, several derivations and variables lack the rigorous prerequisite assumptions and detailed process explanations necessary for a fully exhaustive verification.
Presentation:
The overall structure of the manuscript is logical and adheres to the standard conventions of the field, progressing from a clear motivation in Geometric Deep Learning (GDL) to rigorous theoretical proofs and diverse empirical evaluations. However, while the technical narrative is easy to follow for an expert reader, the submission requires significant linguistic proofreading to address recurring grammatical errors and typographical oversights that detract from its professional presentation.
Significance:
The manuscript addresses a critical and timely problem in geometric deep learning: the effective processing of vector-valued features on non-Euclidean domains. While scalar-based Graph Neural Networks (GNNs) have matured significantly, extending these architectures to handle directional or tangent-bundle data (e.g., wind fields, neural activity, manifold-aligned vectors) remains a non-trivial challenge. By introducing Vector Diffusion Wavelets (VDWs), the authors provide a mathematical bridge between classical diffusion wavelet theory and modern geometric graph learning.
Originality:
The manuscript presents a significant conceptual and technical advancement in the field of Geometric Deep Learning (GDL). By extending the established principles of scalar diffusion wavelets to vector-valued data residing on tangent bundles, the authors address a non-trivial challenge: processing complex directional signals while maintaining rigorous geometric consistency.

---

> ### Author Rebuttal · Authors · 2026-03-29
>
> Re: proofreading — We thank the reviewer for their careful reading and apologize for our typos. We will make sure to fix these errors in the camera-ready copy.
>
> Re: stability under sparsity — Our method does require sufficiently well-sampled local estimates ($n_i \geq D$, as noted in Footnote 2). Otherwise, if the number of neighbors for a node is less than the vector feature dimension, then U will be rank-deficient, and we can’t recover a full basis for that node’s tangent space via local PCA. However, if the data is sparse only in particular regions, one could readily modify the construction of $\mathbf{Q}$ to down-weight the influence of their nodes. (This did not end up being necessary for the datasets used in our experiments.)
>
> Additionally, one could include more points beyond immediate neighbors. For instance, one could use all points $v_j$ where $\|v_j-v_i\|_2\leq \delta\_i$ where $\delta\_i$ is a threshold that accounts for sampling density, analogous to the CkNN graphs considered in [2]. Finally, one could use robust tangent space estimators such as those introduced in [3] or [4]. We will add discussion of these points in the camera-ready version.
>
> Re: self-adjoint operator — The vector diffusion operator $\mathbf{Q}$ is not self-adjoint. However this is not necessary for our proofs.
>
> For Theorem 3.1 (the frame bound), the keys to the proof are as follows. In Lemma B.1, we recall Proposition 2.2 of [1], which shows that the scalar diffusion wavelets $\mathcal{W}\_J$, defined in terms of the scalar diffusion matrix $\mathbf{P}$, are a non-expansive frame on a weighted $\ell^2$ space where $\mathbf{P}$ is self-adjoint (because $\mathbf{P}$ is similar to a symmetric matrix, even though it’s not symmetric itself). We then prove Corollary B.2 to show that these same scalar wavelets are a frame, with frame bounds defined in terms of the vertex degrees, on the standard unweighted $\ell^2$ space. This allows us to establish Lemma A.1 which proves a frame bound for a modified version of the vector diffusion wavelets defined in terms of the $nd \times nd$ matrix $\mathbf{Q}’$ defined in block form by $\mathbf{Q}’[i,j]=\mathbf{P}[i,j] I$ rather than $\mathbf{Q}[i,j]=\mathbf{P}[i,j] \mathcal{O}\_{i,j}$. We then use this lemma, along with Lemma A.2, to complete the proof.
>
> For Theorem 3.2 (wavelet equivariance), the key to the proof is Proposition B.3 which shows $\overline{\mathcal{O}\_{i,j}}=R \mathcal{O}\_{i,j} R^T$. This allows us to establish the equivariance of $\mathbf{Q}$, and its powers in Lemma A.3. Theorem 3.2 follows by linearity. The proof of Theorem 3.3 (scattering equivariance) relies on an induction argument using Theorem 3.2 and the assumption that $\sigma$ commutes with rotations.
>
> Re: oversmoothing — Wavelets-based methods have been shown to help with oversmoothing in prior work, such as [5] (see Table 5 and surrounding discussion) and [6] (Table 1 and surrounding discussion). For VDWs, comparison to standard message-passing GNNs may be difficult. These GNNs treat vector-valued signals as scalars, which may lead to decreased performance with or without oversmoothing. However, as noted by reviewer LU4t, there are also other methods for mitigating oversmoothing (e.g., residual connections, used in comparison methods EGNN and TFN). We will clarify our discussion of oversmoothing in the camera-ready copy.
>
> Re: scalability — We agree that testing VDWs on bigger data sets with additional geometric complexity is an important avenue of future work to confirm their efficacy and parameter efficiency at larger scales. At scale, we note that VDWs can be computed once and applied as a feature extraction step, and the resulting scattering coefficients cached. The computational bottleneck is then computing the local frames via SVD for every node on a graph. When the number of nodes (and/or neighbors) is large, this step is readily parallelized, so our method should scale to larger graphs. Alternatively, a graph may have a vector feature of large dimension, which adds to the expense of the SVDs. In this case, truncated SVDs can be an efficient option, where dimensionality reduction of the vector feature helps make a problem more tractable.
>
> [1] M. Perlmutter et al. Understanding graph neural networks with generalized geometric scattering transforms. SIAM J. on Math. of Data Sci. 2023.
>
> [2] T. Berry & T. Sauer. Consistent manifold representation for topological data analysis. Found. Data Sci. 2019.
>
> [3] D. Kohli et al. Robust tangent space estimation via Laplacian eigenvector gradient orthogonalization. https://arxiv.org/pdf/2510.02308. 2025.
>
> [4] Y. Zhan & J. Yin. Robust local tangent space alignment via iterative weighted PCA. Neurocomputing 2011.
>
> [5] S. Viswanath et al. HiPoNet: A Multi-View Simplicial Complex Network for High Dimensional Point-Cloud and Single-Cell Data. NeurIPS 2025.
>
> [6] Y. Min et al. Scattering GCN: Overcoming oversmoothness in graph convolutional networks. NeurIPS 2020.

---

### Official Review · Reviewer_jdRM · 2026-03-13

**Soundness:** 4
**Presentation:** 4
**Significance:** 3
**Originality:** 3
**Overall Recommendation:** 5
**Confidence:** 4

**Summary:**

This paper generalizes diffusion wavelets for vector-valued functions on graphs, structured in a way to approximate the tangent bundle of a manifold and relevant operators on it. In particular, a vector diffusion matrix is used as an analogue to the connection Laplacian, after which the usual properties of equivariance and stability are developed for use in GNNs.

**Compliance With Llm Reviewing Policy:**

Affirmed.

**Key Questions For Authors:**

1. Addressing the noted weakness: it is clear that your method for constructing $Q$ is based on local PCA. In your experiments, how stable are the results to the choice of $k$ for the construction of the $kNN$ graph used to build $Q$? Moreover, do you have any insights on how well this converges to the "true manifold" as the number of sample points grows?

**Limitations:**

yes

**Strengths And Weaknesses:**

Strengths:

This paper is well-written, and is properly linked to the prior literature on the subject. The method put forth by the authors is reasonable, and is demonstrated to have all of the usual equivariance properties that one would desire from such a method. Moreover, the method is shown to perform well on a variety of tasks.

Weaknesses:

The procedure for constructing the "vector diffusion matrix" $Q$ seems reasonable, but ad-hoc.

---

> ### Author Rebuttal · Authors · 2026-03-29
>
> Re: choice of $k$ — We thank reviewer jdRM for this perceptive question. First, we note that we use $k$-NN graphs in our experiments to induce sparsity and increase computational efficiency and scalability, prioritizing small $k$ where possible, and challenging all models to work with sparse graphs.
>
> Overall, the stability of results to the choice of the hyperparameter $k$ varies by learning task for our method. This is expected, since $k$ helps tune vector feature aggregation across nodes (by controlling how many neighbors’ signals are averaged in each diffusion step) and different degrees of local/rough versus global/smooth signal aggregation can be important to different tasks.
>
> Importantly, the choice of $k$ should be considered jointly with the choice and parameterization of an edge-weighting kernel, if used. For instance, a large $k$ paired with a rapidly decaying kernel such as a Gaussian with a small bandwidth yields a nominally denser graph, but with distant neighbors receiving near-zero weights and contributing little to products involving $\mathbf{Q}$. In this scenario, results would be more dependent on the choice of kernel bandwidth than the choice of $k$ (though computational cost would increase).
>
> The sensitivity of results to $k$ also depends on which spatial scales are most informative for the task. $\mathbf{Q}$ is a smoothing operator, and our wavelets are constructed from powers of $\mathbf{Q}$, capturing differences in vector-valued signals as averaged over various spatial scales. This means $k$ helps control the receptive field of each wavelet in the ambient space $\mathbb{R}^D$: small $k$ tends toward more local receptive fields and slower smoothing, while large $k$ tends toward broader receptive fields and more global averaging. Crucially, as long as $k$ isn't too large, the wavelet filter bank already provides a local-to-global representation of the signal; moderate variation in $k$ shifts which scales are emphasized, but doesn't catastrophically alter what the model sees. This is one advantage of the wavelet approach over multi-layer GNNs: multiscale coverage without large parameter counts.
>
> This idea guided our specific choices of $k$ in each experiment. For the ellipsoid diameter estimation task, global structure was the more important signal, and $k = 5$ was chosen alongside $t_J = 16$ (the maximal diffusion step), yielding sufficiently smooth and global representations for graphs of $n = 128$ nodes. For wind vector reconstruction, where interpolated vectors are most similar to immediate neighbors' known vectors, the hyperlocal signal was most important and $k = 3$ performed best. For the multi-channel neural recording experiments, we follow [1] in using the CkNN graph construction from [2], This approach allows k to vary by point using local density estimation, in order to avoid the potential pitfall of a standard $k$-NN graph connecting distant points when regions are sparsely sampled. (This more complex approach was not necessary for the known ellipsoid and sphere manifolds of the other experiments.)
>
> Re: convergence — As noted in Section 2.2, we view powers of the vector diffusion operator $\mathbf{Q}$ as a computationally efficient proxy for the heat semigroup $e^{-t\Delta}$ associated with the connection Laplacian $\Delta$. The convergence of networks constructed from this heat semigroup was analyzed in [3] (Theorem 1) and thus our network is guaranteed to converge (in probability) if we use a modified family of wavelets defined via the heat-semigroup, e.g., replace $\mathbf{Q}^{2^{j-1}}-\mathbf{Q}^{2^j}$ with $e^{-2^{j-1}\Delta}-e^{-2^j\Delta}$ (defined in the spectral domain). However, in our implementation we prefer to use wavelets defined as powers of $\mathbf{Q}$ since it allows the wavelet transform to be computed via sparse matrix-vector multiplications and thereby increases the computational efficiency of our method.
>
> We will add a remark to clarify this point and also discuss relevant literature on the convergence of the vector diffusion operator [4] and on how $k$ can be chosen to ensure the convergence of manifold neural networks [5].
>
> [1] A. Gosztolai et al. MARBLE: Interpretable representations of neural population dynamics using geometric deep learning. Nature Methods 2025.
>
> [2] T. Berry & T. Sauer. Consistent manifold representation for topological data analysis. Found. Data Sci. 2019.
>
> [3] C. Battiloro et al. Tangent bundle convolutional learning: From manifolds to cellular sheaves and back. IEEE Trans. on Signal Proc. 2024.
>
> [4] A. Singer & H. T. Wu. Spectral convergence of the connection Laplacian from random samples. Information and Inference 2017.
>
> [5] Z. Wang, et al. Generalization of geometric graph neural networks with Lipschitz loss functions. IEEE Trans. on Signal Proc. 2025.

---

> > ### Author Rebuttal · Reviewer_jdRM · 2026-04-04
> >
> > Thank you for the rebuttal. I think my score from before is suitable, so I will keep it as is.

---

### Decision · Program_Chairs · 2026-04-30

**Decision:**

Accept (regular)

**Comment:**

The paper introduces vector diffusion wavelets and integrates them into geometric GNNs for vector-valued data.

Reviewers agree that the approach is technically sound, well motivated, and supported by both theoretical analysis and empirical results, with strong performance across multiple tasks.

While some concerns were raised regarding presentation clarity, assumptions in the construction, and limited large-scale evaluation, the rebuttal addressed these points satisfactorily and clarified the method and its scope.